



# Optimising Ensemble Streamflow Predictions with Bias-Correction and Data Assimilation Techniques

Maliko Tanguy[1,2], Michael Eastman[1,3], Amulya Chevuturi[1], Eugene Magee[1], Elizabeth Cooper[1], Robert H B Johnson[1], Katie Facer-Childs[1], Jamie Hannaford[1,4]

5  [1] UK Centre for Ecology and Hydrology (UKCEH), Wallingford, UK
[2] European Centre for Medium-Range Weather Forecasts (ECMWF), Reading, UK
[3] Met Office, Exeter, UK
[4] Irish Climate Analysis and Research UnitS (ICARUS), Maynooth University, Maynooth, Ireland

*Correspondence to*: Maliko Tanguy (malngu@ceh.ac.uk)

10  **Abstract.** This study evaluates the efficacy of bias-correction (BC) and data assimilation (DA) techniques in refining hydrological model predictions. Both approaches are routinely used to enhance hydrological forecasts, yet there have been no studies that have systematically compared their utility. We focus on the application of these techniques to improve operational river flow forecasts in a diverse dataset of 316 catchments in the UK, using the Ensemble Streamflow Prediction (ESP) method applied to the GR4J hydrological model. This framework is used in operational seasonal forecasting, providing a suitable testbed for method application. Assessing the impacts of these two approaches on model performance and forecast skill, we find that BC yields substantial and generalised improvements by rectifying errors post-simulation. Conversely, DA, adjusting model states at the start of the forecast period, provides more subtle enhancements, with the biggest effects seen at short lead times in catchments impacted by snow accumulation/melting processes in winter and spring, and catchments with high Base Flow Index (BFI) during summer months. The choice between BC and DA involves trade-offs, considering conceptual differences, computational demands, and uncertainty handling. Our findings emphasise the need for selective application based on specific scenarios and user requirements. This underscores the potential for developing a selective system (e.g., decision-tree) to refine forecasts effectively and deliver user-friendly hydrological predictions. While further work is required to enable implementation, this research contributes insights into the relative strengths and weaknesses of these forecast enhancement methods. These could find application in other forecasting systems, aiding the refinement of hydrological forecasts and meeting the demand for reliable information by end-users.

**Key words**

Hydrological forecasts, data assimilation, particle filter, bias-correction, quantile mapping, skill evaluation, Ensemble Streamflow Predictions, seasonal forecasting.



# 1 Introduction

**Table 1: Glossary of acronyms commonly used in this study.**

| Acronym | Meaning | Definition/Comments |
|---------|---------|---------------------|
| BC | Bias Correction | Technique to adjust model outputs to account for systematic errors or biases. |
| BFI | Base Flow Index | A measure of the proportion of the river runoff that derives from stored sources (catchment with long hydrological memory have a high BFI). |
| CRPSS | Continuous Ranked Probability Skill Score | Forecast skill score used in this study. |
| DA | Data Assimilation | Methodology which integrates observed data into models to improve the accuracy of predictions by updating model states and parameters. |
| ESP<br><br>▪ OR-ESP<br>▪ BC-ESP<br>▪ DA-ESP | Ensemble Streamflow Prediction<br><br>▪ Original-ESP<br>▪ Bias-Corrected ESP<br>▪ ESP with Data Assimilation | Streamflow forecasting method used in this study, involving the use of a hydrological model driven by an ensemble of historical climate data to generate probabilistic streamflow forecasts. |
| FDC | Flow Duration Curve | Provides the distribution of flow rates used to apply quantile mapping bias correction method. |
| HOUK | Hydrological Outlook UK | Operational service providing seasonal forecasts and assessments of future hydrological conditions across the United Kingdom. |
| IHC | Initial Hydrological Condition | The state variables of the hydrological system at the beginning of the forecasting period (i.e. initial values of the model parameters). |
| PF | Particle Filter | DA method for estimating the state of a system by representing the probability distribution with a set of samples (particles) that evolve over time based on observations and model dynamics. |
| QM | Quantile Mapping | BC technique to correct model biases by aligning the quantiles of model output with observed data. |

Hydrological forecasts are a critical tool for water resources management, flood forecasting, and drought mitigation. In a warming world, we expect to see an increase in both high and low flow extremes, which will cause a wide range of impacts for society and the environment (Kreibich et al., 2022). Therefore, the need for reliable hydrological forecasts is more critical than ever, such that proactive action to can be taken to mitigate these impacts.



There are different approaches to operational hydrological forecasting, ranging from process-based models to fully data-driven approaches. In the United Kingdom (UK), the Hydrological Outlook UK (HOUK) provides operational forecasts which are used by a range of stakeholders supporting their decision-making (Hannaford et al., 2019). The HOUK uses three different approaches to produce its forecasts (Prudhomme et al., 2017). For the first approach, hydrological models are driven with seasonal weather forecasts produced by the Met Office to derive river flow forecasts (Bell et al., 2013). A second, dual approach, which is purely data driven and based on statistical methods, generates 'persistence' forecasts using flow anomaly in the most recent month as well as 'historical analogue' forecasts using the most similar historical sequences (Svensson, 2016). The third approach, which is the approach we are focusing on in this paper, is the Ensemble Streamflow Prediction (ESP) where a hydrological model is driven by an ensemble of historical climate time series to generate probabilistic streamflow forecasts (Harrigan et al., 2018).

Alternative approaches include long-term average scenarios, where catchment hydrological models are driven by rainfall scenarios assuming specific percentages of long-term average rainfall (e.g., 60%, 80%, or 100%). This method is used in monthly water situation reports by the Environment Agency (e.g., Environment Agency, 2022). Additionally, emerging approaches like the use of storylines and large ensembles to explore plausible worst-case scenarios for upcoming months are gaining popularity in water resources management (e.g., Chan et al., 2024; Kay et al., 2024).

This study will focus on enhancing hydrological predictions using the ESP method, which has long been used worldwide and forms the basis for many operational seasonal forecasting systems (Wood, 2016). The UK provides a testbed for application given the existence of the operational HOUK, but the results of this study could resonate in many other settings. The ESP method, as utilised in this paper, employs historical sequences of climate data (precipitation and potential evapotranspiration) to drive hydrological models, generating a range of possible future streamflow conditions. The source of forecasting skill of ESP method stems from accurate estimation of the initial hydrologic conditions (IHCs), which, depending on the model, can include antecedent stores of soil moisture, groundwater, snowpack, and channel streamflow (Wood, 2016; Wood and Lettenmaier, 2008), rather than from skilful atmospheric forecasts. The ESP therefore offers an ideal environment for testing forecast enhancement techniques since it isolates the skill associated with Initial Hydrological Conditions (IHCs) from that stemming from accurate meteorological forcings. The IHCs can be detected up to a year in advance (Staudinger and Seibert, 2014) depending on catchments characteristics. Harrigan et al. (2018) have shown that in the UK, ESP is particularly skilful in catchments with a long 'memory' due to their strong groundwater influence. These catchments are concentrated in the South-East of the country, where ESP shows forecasting skill for lead times of up to 6 months. In the North-West of the country however, the skill of ESP is limited. This part of the country is dominated by 'flashy' fast-responding catchments, with steeper orography and little groundwater storage, where the IHCs have less predictive power and highlights the limitations of ESP method. Despite its simplicity, ESP outperforms other hydrological forecasting approaches in many cases, and remains a hard-to-beat reference, both in terms of skill but also in terms of value (Peñuela et al., 2020).

The ability of ESP to produce skilful forecasts, as with any model-based forecasting approach, is also inherently linked to the capability of the hydrological models used to produce accurate streamflow simulations. Streamflow simulations produced by



hydrological models contain multiple sources of uncertainties, including the model structure, parameterisation, forcing data, and initial conditions (Renard et al., 2010).

GR4J[1] (Génie Rural à 4 paramètres Journalier) hydrological model was used in this study, and has been shown to reliably simulate the hydrology of a diverse set of catchments (Perrin et al., 2003) including temporal transition between wet and dry periods (Broderick et al., 2016). Smith et al. (2019) demonstrated the good performance of the GR4J model over 303 UK catchments, enabling historic streamflow data reconstruction. However, GR4J is a simple lumped catchment, with only four parameters: (i) a soil moisture accounting reservoir, (ii) a water exchange function, (iii) a non-linear routing store to represent
baseflow, (iv) rainfall-runoff time lags controlled by two-unit hydrographs. This simple model has the advantage of being very quick to run, and computationally inexpensive, which is an essential criterion for an operational service, but might not be able to capture the complexity of some of the hydrological systems, resulting in some biases, particularly towards the extremes.

While Lane et al. (2019) did not include GR4J in their study, they demonstrated a common challenge in hydrological modelling: systematic biases, particularly evident in regions with inadequate snowpack simulation, inter-catchment
groundwater exchange, or significant human influence on the basin. Figure 1a illustrates the scale of model bias for GR4J. Figures A4 and A5 show that the bias is generally greater for low flows when measured as percent bias, whereas it is greater for high flows when considering raw bias values. To address this issue of hydrological model biases impacting on predictions, researchers have developed various approaches to refine forecasts. Two prominent techniques are bias-correction (BC; e.g., Bum Kim et al., 2021) and data assimilation (DA; e.g., Piazzi et al., 2021). While both methods share the common goal of
enhancing forecast accuracy, they diverge fundamentally in approach. BC is a statistically-based post-processing step that adjusts the forecast based on past performance, whereas DA improves the IHCs and corrects internal model states. This fundamental difference may explain why there has been no prior attempt to compare the efficacy of these approaches in operational settings. However, from a user perspective, where the emphasis lies on the reliability of the final product to aid decision-making, such a comparison holds significant value. Ultimately, it can lead to the creation of more reliable end-
products for users.

Several previous studies have shown the advantages of using BC as a post-processing technique to enhance the skill of hydrological forecasts (e.g., Chevuturi et al., 2023; Tiwari et al., 2022). Some operational systems, such as the GEOGloWS ECMWF Streamflow Service, apply BC to generate their forecasts (Sanchez Lozano et al., 2021). Hashino et al. (2007) conducted a study in which they compared various BC methods for ensemble streamflow forecasts, and found that the quantile
mapping (QM) method outperformed other techniques, resulting in a significant improvement in forecast skill. QM stands out

---

[1] When this study was carried out, GR4J (Génie Rural à 4 paramètres Journalier) model (Perrin et al., 2003) was used in the HOUK to produce the ESP forecasts, having been used operationally for five years. This has since then been updated to GR6J (Génie Rural à 6 paramètres Journalier, Pushpalatha et al., 2011) in October 2023. However, the difference in skill between the two model structures is minimal (Appendix, Figure A1, A2 and A3), except for some catchments in the South-East for short lead-time. The mean difference in skill (measured by CRPSS) between the two models is less than 0.03. In this study, we used GR4J, but given the marginal discrepancy in performance between the two models, we anticipate that the findings and conclusions of this study would remain largely applicable when employing GR6J.





as the most frequently employed approach in prior studies using bias-correction for improving streamflow simulations (e.g., Chevuturi et al., 2023; Farmer et al., 2018; Usman et al., 2022). While some researchers opt to bias-correct precipitation and temperature prior to input into hydrological models, Tiwari et al. (2022) found that directly bias-correcting streamflow leads to superior results. Li et al. (2017) presents a comprehensive review of forecast post-processing methods. QM stands out as

one the most popular options in hydrological forecast, due to its simplicity and efficiency (e.g., Hashino et al., 2007; Wood and Schaake, 2008). However, as an unconditional method, QM uses the cumulative distribution function to perform the correction, and so does not preserve the connection between each pair of simulated and observed values. Thus, QM may adjust the raw forecasts in the wrong direction for some forecast values (Madadgar et al., 2014).

Unlike BC, which is applied as a post-processing step, the aim of data assimilation (DA) is to improve the IHCs by combining
models with observed data to improve the estimation of the target variable during the forecast period (e.g., Carrassi et al., 2018). In this way, DA can be seen as an effort to provide a more physically-based improvement of the model predictions, rather than a statistically-based post-hoc correction. DA has a long history of application in meteorological (e.g., Navon, 2009) and hydrological forecasting (e.g., Liu et al., 2012), but in the latter case has tended to be focused on short lead-time (typically the order of days, for flood forecasting applications; e.g., Piazzi et al., 2021). There have been relatively few studies of DA for
sub-seasonal to seasonal forecasts in hydrology.

DA can be performed sequentially, using observed data as it becomes available, to update the model states and/or parameters. In this study, sequential DA of streamflow observations is performed during the model spin-up period to better approximate the IHCs at the start of the forecast period and to update model parameter values. Previous research has demonstrated the potential of sequential DA approaches to improve model performance by reducing initial condition uncertainty (e.g., Piazzi et
al., 2021). Two of the most popular methods are Kalman filters (e.g., Maxwell et al., 2018; Thiboult et al., 2016) and particle filter (e.g., DeChant and Moradkhani, 2011; Jin et al., 2013).

Kalman filter approaches for nonlinear systems, such as the extended Kalman filter, are often limited by their high computational demand, unbounded error growth, and instability in the error covariance equation (Evensen, 1992). Ensemble Kalman filter (EnKf) approaches can be used to overcome some of these issues but rely on the assumption of Gaussian errors
(Evensen, 1994). In contrast, particle filters do not make any assumptions regarding error distributions. However, particle filters may struggle in high-dimensional cases, requiring very large ensemble sizes to avoid 'particle weight collapse', where most particles end up with similar weights, failing to represent the full range of system states (Snyder et al., 2008).

For hydrological forecasts, Piazzi et al. (2021) have shown the potential effect of DA on skill improvement for short lead times (2 days). Other work has shown that the impact of data assimilation and alternative approaches used to improve model skill,
such as precipitation forcing, varies with lead time. However, the majority of research in this area focusses on short- to medium-range forecasts (1-31 day lead-time; e.g., Boucher et al., 2020; Clark et al., 2008; Piazzi et al., 2021; Randrianasolo et al., 2014; Seo et al., 2009; Sun et al., 2015; Thiboult et al., 2016). This is despite the improvements in hydrological forecasting making the production of skilful longer-term forecasts possible (e.g., Harrigan et al., 2018). Only a handful of studies have investigated the impact of initial condition estimates on longer lead times in hydrological forecasts in the United States





(DeChant and Moradkhani, 2011; Shukla and Lettenmaier, 2011), showing generally improved seasonal predictions with DA, but with little added value beyond 1-month forecast. However, beyond this, research into the potential of DA to improve seasonal and sub-seasonal hydrological forecasts' skill is limited. Therefore, there may be potential to improve skill at longer lead times by updating model parameters as well as initial streamflow states.

Note that other approaches, such as multi-model blending, have been used by others to improve forecasts (e.g., Chevuturi et

al., 2023; Roy et al., 2020; Shamseldin, 1997), but will not be considered in this study.

The overall objective of this paper is to evaluate and compare the utility and effectiveness of BC and DA approaches for optimising hydrological forecasts outputs over a range of different lead times. This is achieved through application to a dataset of 316 UK catchments, representing a diverse range of catchment properties. We aim to provide guidance on the relative performance of these methods, and how this varies according to location and catchment type, lead-time and time of year. As

this is based on an operational seasonal forecasting product, the Hydrological Outlook UK ESP forecasts, it will enable users to make informed decisions and will provide insights into the most effective strategies for enhancing UK hydrological forecasting. More generally these results can find application for other hydrological seasonal forecasting systems in other regions, and can underpin future research in improving operational hydrological forecasts.

## 2 Material and methods

### 2.1 Data

#### 2.1.1 River flow data

Observed daily river flow data were obtained for 316 catchments (Figure 1) from the National River Flow Archive (NRFA, https://nrfa.ceh.ac.uk/) database. For the full metadata of these catchments, see supplementary information in Harrigan et al. (2018). The catchment observations were used to calibrate the model (see section 2.2), for bias-correcting the streamflow

simulations (see section 2.3), for model performance evaluation (see section 2.5) and forecast skill assessment (see section 2.6). The study period used was from 1st January 1961 to 31st December 2015.

For the forecast skill assessment, complete observed time series were needed (see section 2.6), so gap-filling, using simple linear interpolation, was applied to the missing data in the observed river flow time series (the limitations of this method are discussed in section 4.4). The gap-filled version of the dataset was only used for the forecast skill assessment, not for the other

applications (model calibration, BC and model performance evaluation).

Considering the amount of missing data in the observational dataset and the diverse hydrological characteristics of catchments, we defined four different subsets of catchments (see Figure 1b):

(i) The full set of catchments (316 catchments);

(ii) Catchments with less than 5% missing observed river flow data (139 catchments);





(iii) Catchments with Base Flow Index (BFI) greater than 0.6 (70 catchments). The BFI is a measure of the proportion of the river runoff that derives from stored sources; the more permeable the rock, superficial deposits and soils in a catchment, the higher the baseflow and the more sustained the river's flow during periods of dry weather (Gustard, 1992). In other words, the higher the BFI, the longer the catchment memory, and therefore improving the IHCs in these catchments has the potential to improve the hydrological forecasts for longer lead times.

(iv) Catchments with BFI greater than 0.6 and missing observed river flow data less than 5% (29 catchments).

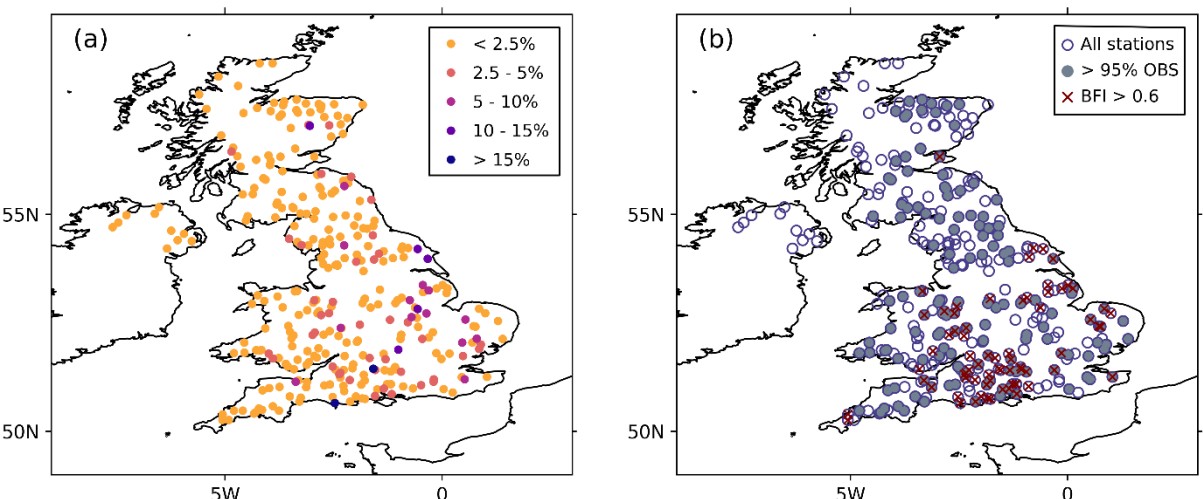

**Figure 1: (a) Absolute percent bias (absPBIAS) in the 316 study catchments for streamflow simulated with GR4J model and (b)**
**location of gauging stations for the 316 NRFA catchments used in this study and their categories based on amount of missing data and value of BFI.**

### 2.1.2 Meteorological data

To run the hydrological model (see section 2.2), precipitation (P) and Potential Evapotranspiration (PET) data is needed. For
P data, we used CEH-GEAR daily rainfall data (Keller et al., 2015; Tanguy et al., 2019). For PET data, we used CHESS-PET (Robinson et al., 2017, 2020) data for Great Britain, and the Historic PET dataset (Tanguy et al., 2017) for Northern Ireland, where CHESS-PET is not available. Tanguy et al. (2018) describe how the Historic PET dataset was derived using a temperature-based PET equation calibrated using CHESS-PET. Consequently, these two datasets can be regarded as almost equivalent, and sufficiently similar for our purposes. The meteorological data used also covered the period from 1961-2015.






## 2.2 Hydrological model, river flow simulations and ESP hindcasts

### 2.2.1 Simulated observed river flows

The hydrological model used to simulate river flow was the GR4J model (Perrin et al., 2003), which served as the operational model for producing ESP forecasts in the HOUK until September 2023. The calibration approach adopted was consistent with
Harrigan et al. (2018), where the modified Kling-Gupta efficiency (KGEmod; Gupta et al., 2009; Kling et al., 2012), applied to root squared transformed flows (KGEmod[sqrt]), was used as the objective function for automatic fitting. This approach places weight evenly across the flow regime, rather than focussing on high or low flows, a decision made considering that ESP forecasts are generated throughout the year, encompassing both dry and wet conditions.

Daily river flow simulations were produced for the period 1st Jan 1964 to 31st December 2015. The initial 3 years (1961-1963)
served as a spin-up period to allow the internal stores to transition from an initial state of unusual conditions to one of equilibrium (Rahman et al., 2016).

### 2.2.2 ESP hindcasts from historical climate

Three versions of ESP hindcasts were used for the period 1964-2014: (i) the hindcasts produced from Harrigan et al. (2018), referred to as 'Original ESP' (OR-ESP) in the rest of the manuscript; (ii) a bias-corrected version of these hindcasts using the
method described in section 2.3, referred to as 'Bias-Corrected ESP' (BC-ESP); and (ii) new hindcasts where the initial conditions at the start of the forecast are corrected using DA method described in 2.4, referred to as 'Data Assimilated ESP' (DA-ESP). Note that what we call 'Original' (OR) is the model simulations with no correction (neither BC nor DA), this is often referred to as 'open-loop' in the literature related to DA (e.g., Boucher et al., 2020).

Each set of hindcasts comprised a 51-member ensemble of streamflow predictions, initiated on the first of each month. These
predictions were generated by forcing GR4J with 51 historic climate sequences (P and PET pairs) extracted for each historic year from 1961 to 2015, projected out to a 12-month lead time at a daily time step. As in Harrigan et al. (2018), to ensure historic climate sequences did not artificially inflate skill (Robertson, 2016), we used a leave-three-years-out cross-validation (L3OCV) approach, whereby the 12-month forecast window and the two succeeding years were not used as climate forcings. This was done to account for persistence from known large-scale climate–streamflow teleconnections such as the North
Atlantic Oscillation with influences lasting from several seasons to years (Dunstone et al., 2016). Each of the 51 generated hindcast time series were then temporally aggregated to provide a forecast of mean streamflow over seamless lead times of 1 day to 12 months, resulting in 365 lead times per forecast (leap days were removed). Following convention in the HOUK, lead time in this paper refers to the streamflow (expressed as mean daily streamflow) over the period from the forecast initialisation date to $n$ days (or months) ahead in time. So, for example, a January ESP forecast with 1-month lead time is the mean daily
streamflow from 1 January to the end of January and a January forecast with 2-month lead time is the mean daily streamflow from 1 January to the end of February.



## 2.3 Bias-correction

The BC methodology applied in this study is a quantile-mapping (QM) approach, similar to that employed by Farmer et al.
(2018). This method was selected for its simplicity of implementation, and its popularity in hydrological applications. QM BC
is applied by Sanchez Lozano et al. (2021) to operationally bias-correct the GEO Global Water Sustainability (GEOGloWS)
streamflow forecasts. Farmer et al. (2018) recommend 14 complete years of observed data to apply this method. This condition
is verified in our dataset, where the shortest record is of 23 years of complete data.

BC was applied separately to each of the 12 months using the observed distribution specific to that month, aiming to capture
seasonality in flow. Figure 2 shows the conceptual diagram of how QM BC works. Each flow value on the simulated Flow
Duration Curve (FDC) is replaced by the flow value of the observed FDC for the corresponding non-exceedance level.

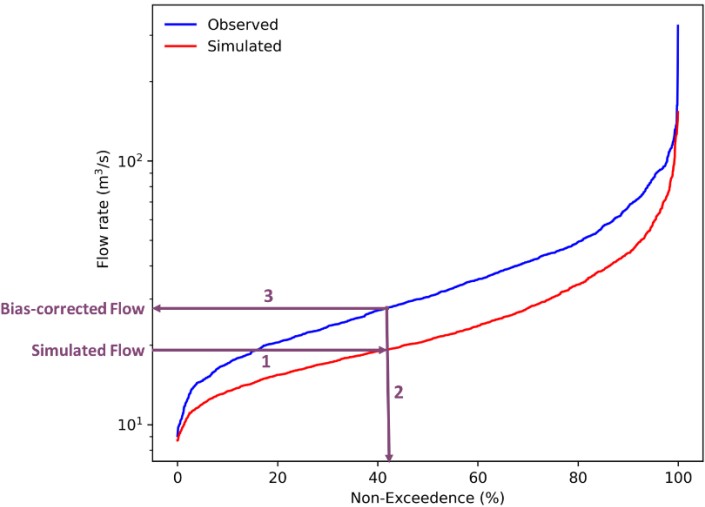

**Figure 2: Conceptual diagram of Quantile Mapping Bias-correction (QM BC), showing the percentage of non-exceedance against**
**the streamflow rate for observed (blue) and simulated (red) streamflow for GR4J over an example catchment 12001 for the month**
**of May. The purple arrows show the steps involved in the bias-correction process: (1) For a given native simulated flow, the point**
**on the simulated Flow Duration Curve (FDC) (red line) is identified; (2) the non-exceedance corresponding to that simulated flow is**
**determined, and (3) the observed flow for that same non-exceedance is determined from the observed FDC (blue line), and this value**
**corresponds to the bias-corrected flow.**


## 2.4 Data assimilation

DA is a group of mathematical methods which can be used to combine information from a numerical model (here a
hydrological model) with available observations to generate an improved estimate of the system's state and, consequently,
more accurate forecasts. DA methods can account for uncertainties associated with model structure, initial conditions, and
observations and provide a probabilistic representation of the hydrological state. Here we used a Particle Filter (PF) technique,



which uses a set of computational particles (representing possible states of the hydrological system) to estimate the most likely current state of the system.

The PF works by simulating multiple potential scenarios (particles) of the hydrological system based on the underlying model but with different sets of model parameters. The method then assigns probabilities to these scenarios based on how well they match the observed data. As new observations become available, the PF updates the particle set, giving more weight to scenarios that align with the most recent data. We updated model parameters (production store, routing store, unit hydrograph 1 level and unit hydrograph 2 level) in GR4J, following the implementation of Piazzi et al. (2021). We used a daily updating timestep in the model spin up period (4 years), in order to improve the IHCs for the seasonal forecast. We chose a PF method over a Kalman filter approach to avoid the restriction of assumed gaussian errors, and so that no mass constraints needed to be applied (see e.g., Piazzi et al., 2021).

## 2.5 Model performance evaluation

To assess model performance, and in particular, compare the improvement provided by BC and DA, we computed a range of performance metrics, detailed in Table 2. These are all metrics commonly used in hydrological assessments (e.g., Hannaford et al., 2023).

**Table 2: List of performance metrics calculated with their corresponding equation. $Q_i$ and $q_i$ are observed and modelled flows for day $i$ of an $n$ daily record. $\overline{Q}$ and $\overline{q}$ are the mean observed and modelled flows.**

| Metric | Abbreviation | Equation | Range and optimum | Focus |
|---|---|---|---|---|
| Root Mean Square Error | rmse | $$rmse = \sqrt{\frac{1}{n}\sum_{i=1}^{n}(Q_i - q_i)^2}$$ | Optimum in 0 (perfect fit). Lower values are better. Range depends on scale of the observations. | Measures the accuracy of the model predictions. |
| Pearson's correlation | Correlation or $r$ | $$r = \frac{\sum(Q_i - \overline{Q})(q_i - \overline{q})}{\sqrt{\sum(Q_i - \overline{Q})^2 \sum(q_i - \overline{q})^2}}$$ | Ranges from -1 to 1, where 1 indicates a perfect positive correlation, -1 indicates a perfect negative correlation, and 0 indicates no correlation. | Measures the linear relationship between observed (O) and predicted (P) values. |
| Bias Ratio | bias or $\beta$ | $$\beta = \frac{\mu_{\sqrt{q}}}{\mu_{\sqrt{Q}}}$$ $\mu$ is the mean flow (here the square root of the modelled and observed flows as indicated by the suffix). | Can be positive or negative. Optimum is 0, indicating no bias. | Measures the systematic overestimation or underestimation of the model. |
| Mean Absolute Percent Error | MAPE | $$MAPE = \left(\frac{1}{n}\sum_{i=1}^{n}\left|\frac{Q_i - q_i}{Q_i}\right|\right)100$$ | Percentage values, lower values are better. Optimum is | Measures the average percentage difference between observed ($Q_i$) and |





| | | | 0, indicating a perfect fit. | predicted ($q_i$) values. |
|---|---|---|---|---|
| Modified Kling-Gupta Efficiency | KGE2 | $KGE2 = 1 - \sqrt{(r-1)^2 + (\beta - 1)^2 + (\gamma - 1)^2}$ where $r$ is the correlation coefficient, $\beta$ is the bias ratio, and $\gamma$ is the variability ratio $\frac{cv_{\sqrt{q}}}{cv_{\sqrt{Q}}}$ or $\frac{\sigma_{\sqrt{q}}/\mu_{\sqrt{q}}}{\sigma_{\sqrt{Q}}/\mu_{\sqrt{Q}}}$ $\mu$, $\sigma$ and $CV$ are the mean, standard deviation and coefficient of variation of flow (here the square root of the modelled and observed flows as indicated by the suffix). | Ranges from -∞ to 1. Higher values are better, with 1 indicating a perfect fit. | Comprehensive metric considering correlation, variability, and bias. |
| Absolute Percent Error in Q95 (flow exceeded 95% of the time) | Q95_APE | $Q95_{APE} = \left\lvert \frac{Q95 - q95}{Q95} \right\rvert 100$ $Q95$ and $q95$ are the 95th percent exceedance for the observed and modelled flow (or 5th percentile). | Percentage values, lower values are better. Optimum is 0, indicating a perfect fit. | Specifically targets errors in predicting low-flow events. |
| Absolute Percent Error in Q05 (flow exceeded 5% of the time) | Q05_APE | $Q05_{APE} = \left\lvert \frac{Q05 - q05}{Q05} \right\rvert 100$ $Q05$ and $q05$ are the 5th percent exceedance for the observed and modelled flow (or 95th percentile). | Percentage values, lower values are better. Optimum is 0, indicating a perfect fit. | Specifically targets errors in predicting high-flow events. |


### 2.6 Forecasts skill assessment

Forecast skill refers to the relative accuracy of a set of forecasts, with respect to some set of standard reference forecasts (Wilks, 2019). Even if the model performance metrics (presented in 2.5) improve with BC and DA, it is not necessarily going to translate into direct improvement in forecasting skill. This is because the enhancement achieved through DA focuses in
improving IHCs, whose impact decays over lead time. Conversely, while BC is expected to enhance simulations across all lead times, its effectiveness is constrained by the inherent limitations linked to the lack of skill in the meteorological forcings, particularly in the case of ESP which relies on climatological data.

The Continuous Ranked Probability Skill Score (CRPSS; Hersbach, 2000) was used in our study for evaluating the probabilistic skill of OR-ESP, DA-ESP and BC-ESP, using climatology as our reference forecast like in Harrigan et al. (2018).
CRPSS penalizes biased forecasts and those with low sharpness (Wilks, 2019). The Ferro et al. (2008) ensemble size correction for CRPS was applied to account for differences between the number of members in the hindcasts (51 members, corresponding to the historic period from 1961-2015 with L3OCV approach) and the benchmark (47 members, corresponding to the period of 1965-2015 with L3OCV approach and four years were removed for the spin-up period), as done in evaluation of hydrological ensemble forecasting elsewhere (e.g., Crochemore et al., 2017). Calculation of skill scores was undertaken using
the open source "easyVerification" package v0.4.2 in R (MeteoSwiss, 2017).

Unlike Harrigan et al. (2018), who employed simulated observed river flows as the 'truth' for skill evaluation, our study relies on observed flows. This choice ensures a fair comparison between OR-ESP, DA-ESP and BC-ESP. Considering DA's





objective of using observations to enhance models, using simulated observed data as the reference would have adversely affected the skill assessment of DA.

The performance metrics (Section 2.5) and skill scores (Section 2.6) were calculated for all three versions of the hindcasts (OR-ESP, BC-ESP and DA-ESP).

We have also calculated other skill scores, namely the mean absolute error skill score (MAESS) and the mean square error skill score (MSESS). However, these are deterministic skill scores, and therefore less suited than CRPSS for ensemble forecast verification. Hence, we only show CRPSS results in the following sections for brevity.


## 3 Results

### 3.1 Model performance

BC and DA both improve overall model performance (Figure 3), though in DA the improvement is only marginal and not for all metrics, whereas for BC the difference is more substantial and generalised for all metrics considered.

The greater impact on model performance observed in BC compared to DA is unsurprising given their fundamental differences in their approaches. In BC, observations serve as the absolute 'truth', guiding adjustments to align simulations with the observed Flow Duration Curves (FDCs). As its name implies, BC is explicitly designed to rectify predictions by conforming them to observations, thus naturally yielding improvement in overall performance. Conversely, DA endeavours to enhance predictions through a mechanistic, physically-informed approach during model simulation. In DA, both model-generated and observed values are weighed, aiming to refine the model's alignment with observed data while preserving the hydrological model's structural integrity. Consequently, it is expected that DA may exhibit comparatively lower performance due to the complex interplay between model fidelity and alignment to observed data.





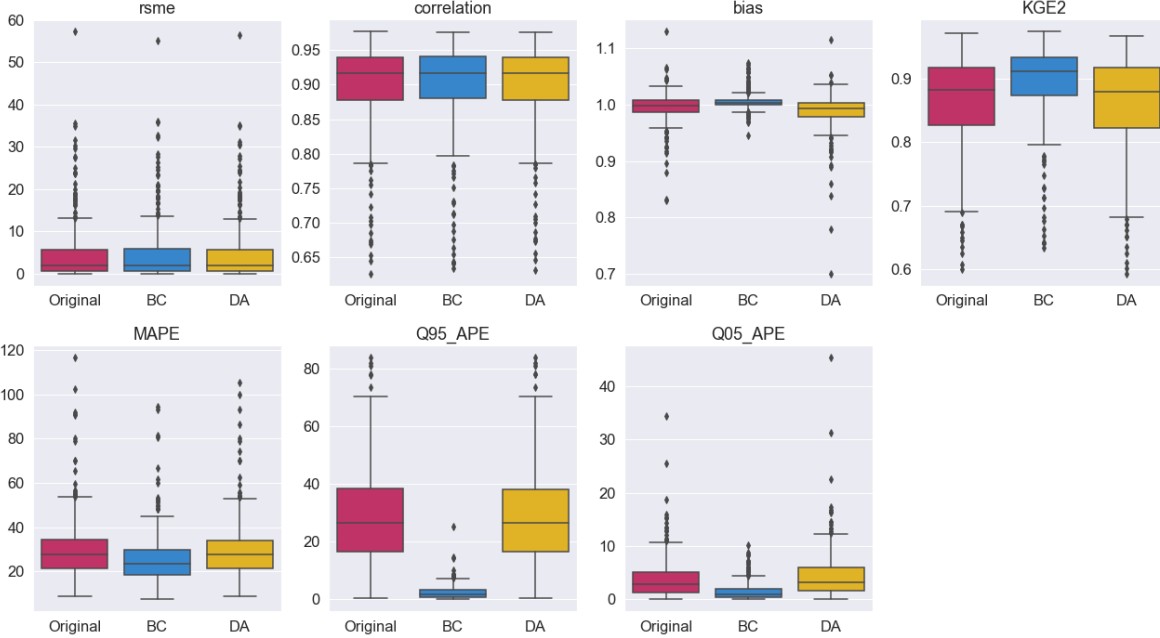

**Figure 3: Model performance metrics for river flow simulations produced by the GR4J model with no additional processing (dark pink), GR4J with BC (blue) and GR4J with DA (orange) for all 316 catchments (Figure1b). The period used to produce the simulations and calculate the metrics is 1964-2015 (1961-1963 used as spin-up period) with observed rainfall and PET as driving data.**

## 3.2 Forecast skill improvement with Data Assimilation

Figure 4a shows the evolution of the skill score (CRPSS) with lead time for OR-ESP and DA-ESP for all catchments with less than 5% missing data. We can see in this figure that overall, there is not a big improvement in skill with DA (no difference in the median skill score). However, the envelope is wider at the top end, especially for very short lead-times, suggesting that DA does make a different for some catchments.

If we look at the same comparison for only catchments with BFI > 0.6 (Figure 4b), the improvement with DA is more notable. This improvement is observed for lead time up to a season (~3 months). After that, the effect of improved initial conditions diminishes.



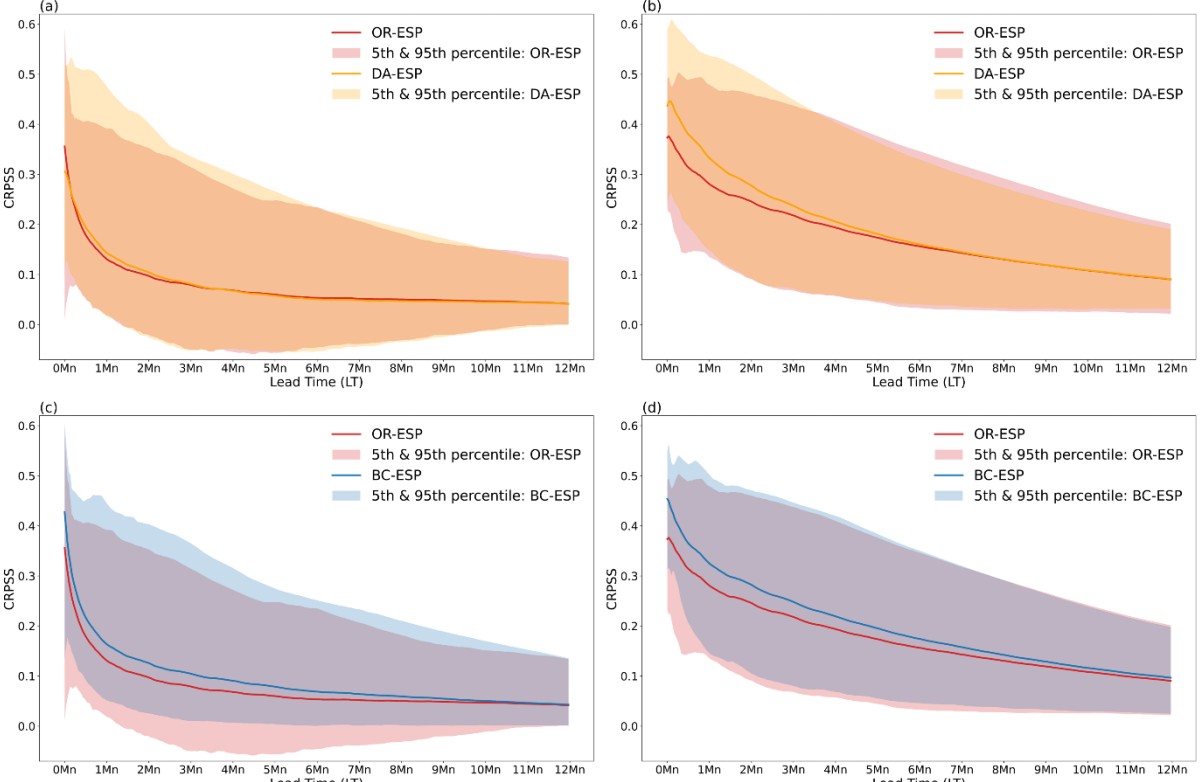

**Figure 4: (a) CRPSS for all stations with <5% missing observations, for OR-ESP (red) and DA-ESP (orange) simulations over lead time; (b) CRPSS for all stations with <5% missing observations and BFI >0.6, for OR-ESP (red) and DA-ESP (orange) simulations over lead time; (c) CRPSS for all stations with <5% missing observations, for OR-ESP (red) and BC-ESP (blue) simulations over lead time; (d) CRPSS for all stations with <5% missing observations and BFI >0.6, for OR-ESP (red) and BC-ESP (blue) simulations over lead time. The solid lines show the median CRPSS, whereas the shaded areas show the catchment spread of 5th-95th percentiles.**

Figure 5a-f shows difference in skill (comparing skill for the OR-ESP on the x-axis against skill for the DA-ESP on the y-axis) in more detail for different types of catchments (low and high BFI) and different lead times. The improvement in skill with DA is more apparent for higher BFI catchments, especially for lead-times 3 to 30 days (as there are more green triangles above the 1:1 line). We also observe that catchments with high Base Flow Index (BFI) exhibit greater overall skill, reflected in higher CRPSS values for both OR-ESP and DA-ESP, which is in line with findings from Harrigan et al. (2018). Figure 5g-l shows results for high BFI catchments only, and with a breakdown between seasons; winter (green triangles) and summer (brown dots), with forecasts initialised the 1st of each month within these seasons. We can see that the improvement brought by DA is much stronger in summer, especially at short lead-times.



**Figure 5: (a to f) Scatter plots of CRPSS between OR-ESP and DA-ESP for all catchments with less than 5% missing observation data, broken down according to the BFI of the station, with low BFI < 0.6 (brown dots) and high BFI > 0.6 (green triangles); (g to l) Scatter plots of CRPSS between OR-ESP and DA-ESP for all stations with <5% missing observations and a BFI greater than 0.6, broken down according to season, December, January, February start months (DJF, winter; green triangle); June, July, August start months (JJA, summer; brown dots); (m to r) Same as a-f but for BC-ESP instead of DA-ESP; (s to x) Same as g-l but for BC-ESP instead of DA-ESP. Forecasts are initialised the 1st of each month. The subplots within each category show increasing lead time (in days).**

## 3.3 Forecast skill improvement with Bias-Correction

In the case of BC, the improvement in skill is longer lasting and more generalised (Figure 4c). Moreover, there is not such a clear difference in improvement between catchments with BFI>0.6 and the rest of the catchments (Figure 4c vs 4d). Notably,



BC improves even the skill of the most poorly performing catchments, as evidenced by the upward shift of the lower bound of the skill envelope (Figure 4c), ensuring that all catchments achieve positive skill scores, a contrast to the performance of DA. Figure 5m-x mirrors Figure 5a-l, but focuses on BC instead of DA, comparing the skill of OR-ESP and BC-ESP across various catchment types (Fig. 5m-r) and seasons (Fig. 5s-x). In this case, no discernible difference in skill improvement between high

and low BFI catchments is evident with BC (Fig. 5m-r). However, we can see that the improvement in skill is greater for catchments with poor original performance. When narrowing our focus to high BFI catchments alone (Fig. 5s-x) and investigating the seasonal effect, we observe that, similarly to DA, skill enhancements are more prominent in summer with BC as well, although to a lesser extent than with DA. This general tendency of better skill in summer is also true for all catchments in the case of BC (not shown).

### 3.4 Data Assimilation vs Bias-Correction

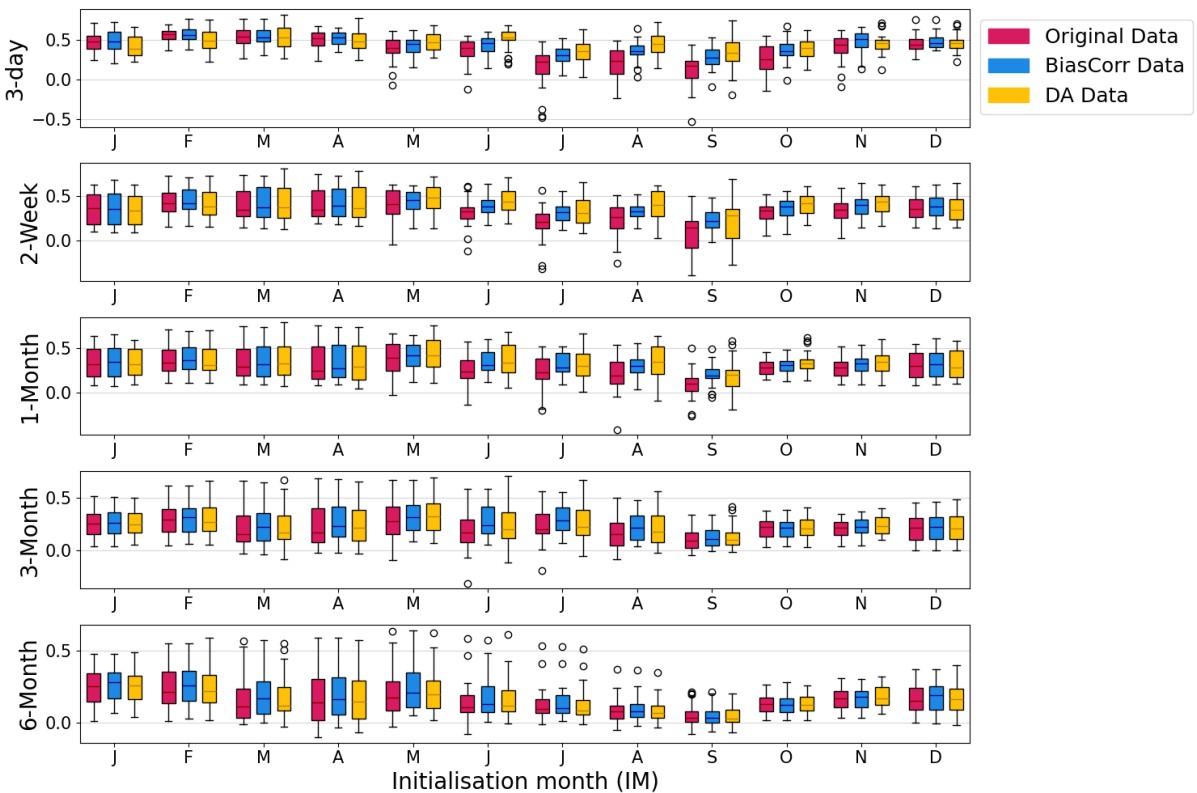

**Figure 6: CRPSS of OR-ESP, BC-ESP and DA-ESP forecasts at different lead times and initialisation months, for catchments with <5% of missing data and BFI>0.6 (the equivalent figure for all catchments with <5% of missing data can be found in the Appendix**
**Figure A6)**



In comparing forecast skills for DA and BC at different lead-times and seasons for catchments with BFI > 0.6 (Figure 6), distinctive patterns emerge: in summer, up to 1-month lead time, DA-ESP outperforms OR-ESP and BC-ESP; whereas BC exhibits higher improvement in skill over winter and at longer lead times. The notable suitability of DA for summer months in high BFI catchments (i.e. with high hydrological memory) underscores the importance of accurate IHCs during drier periods. During such periods, precipitation tends to be closer to climatology, which is what is used to drive ESP. Getting IHCs right through DA in these situations will have a long-lasting effect.

Figure 7 (which displays all 316 catchments) shows spatial differences, notably with DA showing better performance in snow-dominated catchments during winter and spring, especially for short lead times in north-eastern Scotland (Figure 7a). Figure A7 shows the fraction of precipitation falling as snow for catchments across Britain. The version of GR4J used in this study lacks the capability to model snow accumulation and snowmelt processes, making it less reliable in catchments affected by them. DA is especially effective in adjusting the IHCs during seasons influenced by snow, such as winter accumulation and spring melting, when errors in IHCs can be large. However, for longer lead times (Figure 7b), while no distinct patterns emerge, BC generally exhibits better performance across the majority of catchments over most months. This observation suggests a nuanced interplay of factors influencing forecast skill, with BC showing a more consistent advantage in extended lead times across diverse catchment conditions. As mentioned previously, this can be attributed to the fundamental differences in both methodologies.

It is also interesting to note that there are cases where OR-ESP is better than both DA-ESP and BC-ESP (magenta points in Figure 7), especially in autumn, winter and beginning of spring (October to March) in the western part of the country for short lead-times (Figure 7a); and in spring for longer lead-times (Figure 7b) with no clear spatial pattern.



**Figure 7: Each catchment showing of best performing method (OR-ESP: Red; BC-ESP: Blue and DA-ESP: Yellow) for each month at lead time of (a) 3 days and (b) 30 days.**

Figure 8 presents a series of histograms illustrating the differences in CRPSS across various versions of ESP for all 316
catchments, showcasing the extent and variations in improvement offered by BC and DA across different seasons and lead
times. Examining the first two columns of subplots (1st column comparing BC-ESP vs OR-ESP, and 2nd column comparing
DA-ESP vs OR-ESP), we observe some similarities: (i) the range of differences between the corrected (BC or DA) and original

(OR) ESP is narrower in winter (dark blue) and widest in autumn (orange); (ii) spring (light blue) exhibits the most cases
where OR-ESP outperforms both BC-ESP and DA-ESP (indicated by negative values in the histograms); (iii) for both methods
(BC and DA), the greater gains in skill are achieved in summer and autumn; (iv) for lead times longer than 3 months, the
differences between different ESP versions are minimal, with absolute values < 0.08 for BC-ESP vs OR-ESP and < 0.03 for
DA-ESP vs OR-ESP, suggesting negligible improvement beyond this point. Therefore, the gain achieved beyond 3 months

using either technique is marginal.

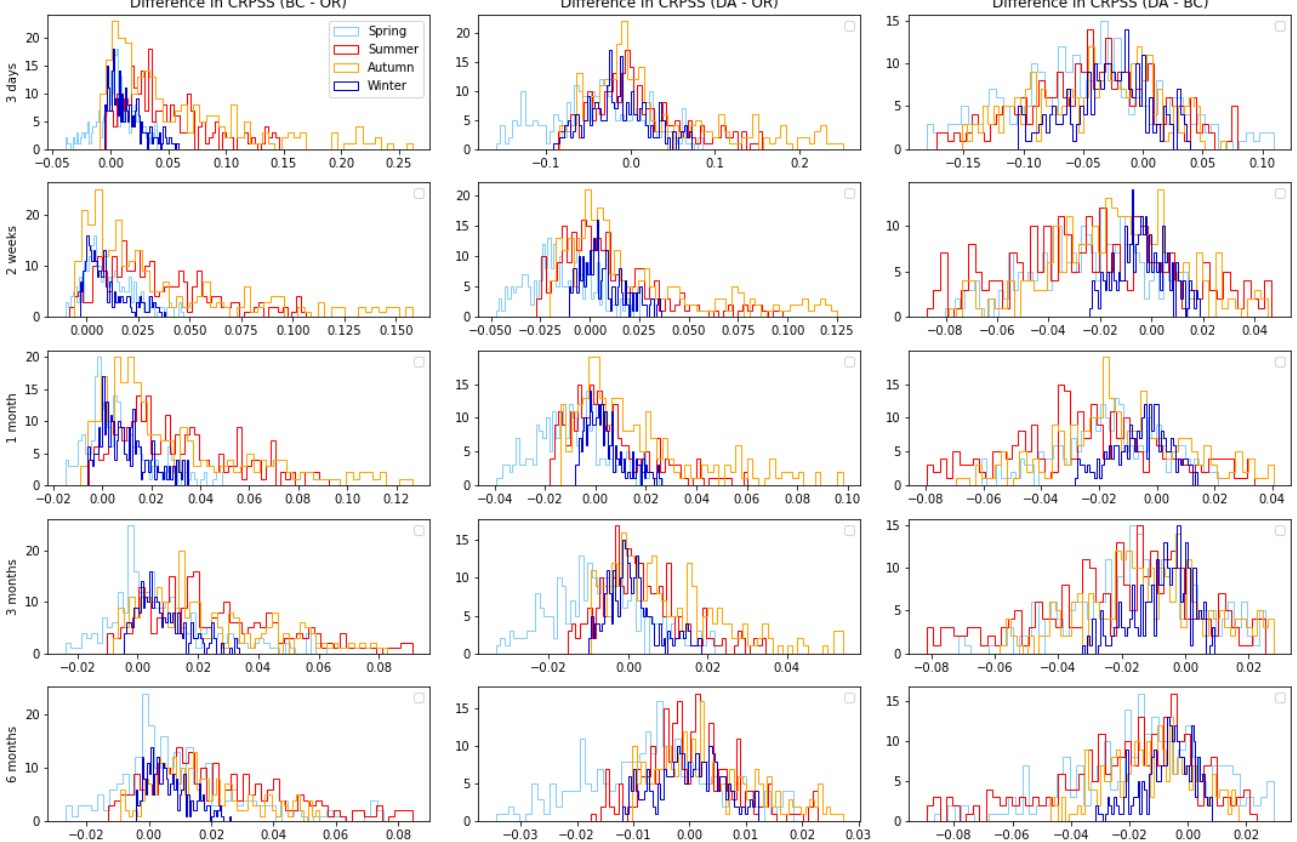

**Figure 8: Histogram showing the difference in CRPSS values across 316 catchments between BC-ESP and OR-ESP (first column),
DA-ESP and OR-ESP (second column), and DA-ESP and BC-ESP (third column), across various lead times (rows) and seasons
(colours). Positive values indicate that the CRPSS value of the first ESP version in each pair is greater than the second (indicating**
**higher skill).**





Focusing now on the differences between DA and BC, we can see that, in general, BC presents fewer negative instances compared to DA, indicating that BC-ESP outperforms OR-ESP more frequently than DA-ESP. However, the magnitudes of improvement are typically comparable for lead times under 3 months for both methods. Directly comparing the CRPSS of 400 DA-ESP and BC-ESP (third column of subplots in Figure 8), we observe a skew towards negative values, indicating more instances where BC outperforms DA. Nonetheless, beyond a 2-week lead time, the absolute differences are negligible (<0.08), suggesting that both methods yield similar outcomes.

## 4 Discussion

### 4.1 Bias-Correction vs Data Assimilation

Despite their shared goal of enhancing forecast accuracy, it is important to recognise the fundamental and conceptual differences between DA and BC methodologies. As already mentioned previously, BC operates as a post-processing technique, rectifying model errors after simulations, while DA intervenes during model initialisations, adjusting model internal states to nudge the simulations towards observed data. DA, as used here, primarily focuses on refining initial conditions, and hence yields more significant impacts in catchments with high BFI due to their extended hydrological memory. DA also proves 410 superior at short lead times for snow-dominated catchments, where the IHCs can be widely wrong due to the lack of explicit representation of snow accumulation and snowmelt processes in the hydrological modelling used in this study. In contrast, BC extends its improvement beyond the initial conditions, improving the quality of simulations throughout the entire time series. However, it is noteworthy that DA and BC methods also fundamentally differ in their handling of uncertainties. DA methodologies based on Bayesian statistics, such as PF, account for uncertainties associated with model structures, initial 415 conditions, and observations, providing a probabilistic representation of the hydrological state. This probabilistic nature enables a more thorough understanding of the forecast, acknowledging the inherent uncertainty in predicting natural systems. Additionally, DA offers the advantage of maintaining the structural integrity of the hydrological model. In other words, a model with a DA-updated initial conditions preserves the relationships between the model state and the target variable, while BC can alter them. Moreover, BC, while effectively aligning model outputs with observations, may inadvertently mask or 420 underestimate the uncertainties in the hydrological model and observational data. The deterministic nature of BC can oversimplify the complex interplay of factors influencing streamflow predictions and uncertainties in observations, potentially leading to an overconfident representation of forecast accuracy.

While the handling of uncertainties distinguishes DA from BC, it is imperative to consider the associated computational demands and implementation complexities, especially if they are to be implemented operationally. This introduces a pragmatic 425 dimension to the comparison, as the choice between DA and BC necessitates a nuanced evaluation of their distinct features and trade-offs. DA's computational demands and implementation complexity starkly contrast with the simplicity and ease of implementation offered by BC, positioning the latter as an accessible 'easy-win' for swiftly enhancing forecasting products. Our results reveal an absence of universal superiority for one method over the other, underscoring their dependency on





catchment characteristics, seasonal dynamics, and lead times. Interestingly, there are cases where the uncorrected OR-ESP
outperforms both DA-ESP and BC-ESP, even at shorter lead times (Figure 7). Therefore, based on our findings, for the UK,
we recommend selectively applying DA for short lead times in summer and catchments with high BFI, and for catchments
affected by snow accumulation/snowmelt in winter and spring at short lead times, which is where the greatest benefit of DA
was observed (Figure 6). In the rest of the cases, BC is the recommended method. Alternatively, the exclusive use of BC is
advocated as a pragmatic, efficient solution, particularly where computational costs pose a limitation. Figure 8 has shown that
even in the rare cases where OR-ESP outperforms BC-ESP, it does so with only marginal differences in CRPSS. This ensures
that the hydrological post-processing is not "doing any harm" to the inherent skill of the raw model outputs (Hopson et al.,
2020).

It should be noted that applying both BC and DA simultaneously to harness their combined effect isn't straightforward. This
complexity arises from the fact that the FDC used for quantile mapping relies on observed flow data without incorporating
data assimilation. Thus, introducing data assimilation would disrupt the established relationships that underpin quantile
mapping method. However, the two methods are not mutually exclusive, and combining them could be evaluated in future
work. This would require a different experimental setting to ensure the applicability of QM BC in simulations that have
undergone data assimilation.

## 4.2 Model Improvements vs Practical needs: A Fine Balance

Many argue that science should focus on the enhancement of hydrological models rather than correcting their errors by post-
processing (BC) or "nudging" their parameters through data assimilation (Refsgaard et al., 2023). While such advancement is
undoubtedly crucial for deepening our comprehension of the physical world (Beven, 2019), this type of research is much
slower to conduct, and incremental improvements take time to translate into impact for users. We could even argue that no
model will ever perfectly model the physical world, e.g., human influence is notoriously difficult to account for in hydrology.
This slow-evolving advancement of the models is asynchronous with the urgent need from end-users to have reliable outputs
that they can trust to base their decision-making on (e.g., Cassagnole et al., 2021; Li et al., 2019; Pappenberger, 2024). It is
worth noting the practical implications of improving hydrological forecasts (Lopez and Haines, 2017; Neumann et al., 2018),
particularly in the context of operational use by decision-makers, whereby the difference between correcting or not correcting
the forecast – regardless of the method being used – can determine whether they are deemed useful and consequently used by
stakeholders or not (Hopson et al., 2020). The Hydrological Outlook UK, which is the subject of this case study, serves as a
pertinent example of the real-world application of forecast products.

Moreover, incremental refinements in hydrological models often result in only marginal enhancements to forecast skill, as
illustrated by plots comparing the GR4J and GR6J models in the Appendix (Figure A1). Notably, the difference in forecasting
skill between these models is minimal, highlighting the challenge of achieving substantial gains through model enhancements
alone. In contrast, our assessment demonstrates that methods like BC yield more significant and immediate improvements in





forecast accuracy. Therefore, in parallel with the ongoing efforts to enhance the core of hydrological models, exploring methods – such as BC and DA – to refine existing forecasting products, becomes justified. This paper consistently assesses and compares two of these methods, offering evidence of the benefits they deliver, and providing a pragmatic solution to refine existing forecasting products, meeting the pressing demand from end-users for reliable outputs that inform their decision-making.

## 4.3 Postprocessing Methods in Hydrological forecasting

In the broader context of hydrological research, our study's exploration of BC and DA methods contributes to the ongoing dialogue surrounding hydrological forecast post-processing techniques. While machine learning (ML) methods like Convolutional Neural Networks (CNN) and Support Vector Regression (SVR) have shown promise in enhancing forecast accuracy (Liu et al., 2022), their computational demands present challenges. Notably, our study employs quantile mapping (QM) for BC, a computationally efficient method, distinguishing it from more resource-intensive ML approaches.

In similar work, Matthews et al. (2022) adopt a post-processing method derived from the Multi-Temporal Model Conditional Processor (MT-MCP; Coccia and Todini, 2011). They found a pronounced impact in catchments with high hydrological memory. A key difference with our work lies in their use of Numerical Weather Prediction (NWP) as driving data, contrasting with our reliance on climatological weather input for the ESP method. This distinction implies that their hydrological forecasts, particularly in dynamic or 'flashy' catchments, are sensitive to the skill of NWP input, potentially diminishing the relative effectiveness of post-processing compared to our experimental setting in those catchments. In that sense, ESP serves as an ideal test case to isolate the value of DA and BC methods, given the absence of skill from the meteorological forecast to "take over" from the IHCs.

Additionally, we acknowledge the challenge of non-stationarity in QM, a concern highlighted by Ceola et al. (2014). To address this, in an operational setting, incorporating real-time updates by dynamically adding the newest observed data would be implemented, allowing the Flow Duration Curve (FDC) to adapt monthly. A potential refinement could be explored by changing reference periods, such as the most recent 30-year period, offering a dynamic approach to account for non-stationarity, albeit with the associated risk of overlooking rarer extreme events.

To the best of our knowledge, no prior study has undertaken a direct comparison between BC and DA, a gap that might be attributed, in part, to the inherent disparities between these two methodologies, as mentioned already. Nevertheless, from a users' standpoint, such a comparative analysis holds significant value. It facilitates the selection of the optimal forecasting product tailored to distinct situations, offering valuable insights for decision-makers seeking to enhance the reliability of their hydrological forecasts.



## 4.4 Limitations and future work

The present study naturally has some limitations given the practicalities of applying multiple approaches across many catchments/lead-times and seasons. Firstly, the gap-filling approach employed to address missing data in the observed river flow time series is quite rudimentary, relying on a simple linear interpolation method. While this method is commonly used for gap-filling (Niedzielski and Halicki, 2023), it comes with inherent limitations, such as its sensitivity to outliers and oversimplification of the underlying hydrological processes, especially when gap-filling longer time periods. Note that both techniques (BC and DA) can be applied even if the observed data has some missing data (DA is not applied where no data is available, and BC uses whatever data is available to construct the FDC). Full time series were only needed to calculate the skill score CRPSS used to carry out the comparative analysis. While more sophisticated techniques for handling missing data, such as data-driven methods or advanced statistical approaches, could have been considered to enhance the accuracy of the reconstructed time series (e.g., Dembélé et al., 2019; Luna et al., 2020), such methods would have significantly increased the complexity of the analysis. To mitigate this limitation, the study has focused much of its analysis on a subset of the dataset, where less than 5% of the data was missing (Figure 1b), minimising in that way the effect of the gap-filling (Arriagada et al., 2021). Consequently, using an alternative gap-filling method would have likely yielded comparable conclusions.

Secondly, the DA methodology implemented in this study, the PF technique, is used in a deterministic manner (where we have used the PF ensemble mean to avoid having an ensemble of ensembles) to ensure comparability with BC results. However, the PF method inherently provides valuable information on uncertainty associated with the hydrological state (e.g., Moradkhani et al., 2012). In the current study, this information on uncertainty is not fully exploited, as the analysis primarily focuses on the comparison with BC. Future investigations could explore more sophisticated approaches within DA that capitalise on the uncertainty estimates provided by PF.

Building on our analysis of the comparative strengths of DA and BC, our study identified the specific scenarios where each method improves the forecasts the most. Depending on the resources available for implementation, we summarise our recommendations based on our findings in Table 3. This lays the groundwork for a prospective user-friendly hydrological forecasting system in the future, which could be implemented in the operational HOUK setting. Recognising that end-users and non-specialists often prioritise a simplified and trustworthy message (Hannaford et al., 2019), we envision the implementation of a flexible, combinatory (e.g., decision tree-based) forecasting system that would dynamically choose the most effective method based on specific factors such as catchment characteristics, time of year, and lead-time. For end-users seeking 'the best answer' without delving into the intricacies of methodology, this streamlined approach aims to provide the most reliable forecast available in a clear and simple manner. While this concept would require rigorous testing and development first, it highlights a potential avenue for future research in tailoring hydrological forecasts to meet the practical needs and expectations of end-users. The cheaper, more immediate solution would be to blanket apply BC to improve all catchments indiscriminately.





**Table 3: 'Best' method and recommended implementation based on available resources.**

| | | OR-ESP | DA-ESP | BC-ESP |
|---|---|---|---|---|
| Where and when is each method recommended? | | For a few cases, OR-ESP outperforms other methods, but generally only marginally compared to BC-ESP. | Best method for the following two cases:<br><br>1) High BFI (>0.6) catchments:<br>▪ Season: summer<br>▪ Lead times: up to 1-month<br><br>2) Catchments where snowpack/snowmelt processes are important:<br>▪ Season: winter and spring<br>▪ Lead time: up to a few days | General improvement for all seasons.<br><br>Lead times:<br>▪ Up to 6 months in summer and autumn.<br>▪ Up to 1-3 months in winter and spring. |
| Recommended implementation depending on available resources for development and implementation<br><br>(UKHO as example operational forecasting system) | | Not used | Not used | Applied everywhere |
| | Limited resources | Development needed: minimal.<br>▪ Plug-in BC code to the end of current UKHO operational workflow;<br>▪ Optionally: implementation of time-varying reference period for constructing FDC.<br><br>Running cost: minimal. The BC is cheap and quick to run. | | |
| | Unlimited resources | Develop a system which will select the 'best' method (OR-ESP or DA-ESP or BC-ESP) for each catchment depending on season and lead time (e.g., decision tree), where the user clicks on their selected catchment and receives a single, user-friendly 'best possible' answer. This might need a 'seamless' toggle option using a single method for all lead times if step-changes are to be avoided.<br><br>Development needed:<br>- Development of the decision tree;<br>- Development of the 'seamless' toggle option;<br>- New design and deployment of web-interface delivering the forecasts, with careful consideration of how to communicate the underlying methods to the users. This would require stakeholders' engagement to make sure the new product is intuitive and understandable by end-users.<br><br>Running cost:<br>- Running DA operationally will have a substantial cost, and might require the use of HPC facilities in order to deliver forecasts in time. | | |

Furthermore, future studies could explore alternative post-processing methods (Li et al., 2017), such as copula-based approaches and machine learning techniques (Liu et al., 2022), or statistical and empirical post-processing methods, such as

the Hydrological Uncertainty Processor (HUP; Krzysztofowicz, 1999) and its variants, such as Model Conditional Processor (MCP, Todini, 2008). All these options, while potentially offering improved forecast accuracy, come with varying computational expenses. Investigations into diverse post-processing methodologies can enhance our understanding of their applicability and effectiveness in different hydrological contexts, providing valuable options for refining forecasting products in the future.



## 5 Conclusion

In this study, we have explored the effectiveness of Quantile Mapping (QM) Bias-Correction (BC) and Particle Filter (PF) data assimilation (DA) techniques in enhancing hydrological model performance and forecast skill, specifically focusing on improving hydrological forecasts using the Ensemble Streamflow Prediction (ESP) method with GR4J model for the Hydrological Outlook UK operational service. Our findings reveal that both BC and DA contribute to improvements, yet their impacts vary across different metrics and catchment characteristics.

BC, operating as a post-processing method, demonstrates substantial and generalised improvements across various performance metrics. It rectifies model errors after simulations, extending its positive influence beyond initial conditions throughout the entire time series. However, while QM BC effectively aligns statistical properties, it may oversimplify the complexity of hydrological systems by neglecting to capture the physical processes and interactions, consequently leading to an underestimation of uncertainties.

On the other hand, DA, which adjusts model internal states during initialisations to align simulations with observed data, exhibits more subtle and marginal improvements. The positive effects of DA are particularly notable in catchments with high Base Flow Index (BFI) and up to the seasonal scale, and DA often yields more improvement than BC at short lead times (up to one month) in summer. DA also outperforms BC for catchments where snow processes are important, mainly in north-eastern Scotland, in winter (snow accumulation) and spring (snowmelt) at short lead times. The probabilistic nature of DA, considering uncertainties associated with model structures, initial conditions, and observations, provides a comprehensive representation of the hydrological state.

The choice between BC and DA involves trade-offs, considering their conceptual differences, computational demands, and handling of uncertainties. While DA offers a more sophisticated approach, BC presents a pragmatic and computationally efficient solution, especially when computational costs pose a limitation. The absence of universal superiority for one method over the other emphasises the importance of selectively applying these techniques based on specific scenarios, user requirements, and operational constraints. Future work could explore the combined use of both techniques, though it would need to first address the challenge of constructing the Flow Duration Curve used in the QM method for flow simulations modified by data assimilation.

In the broader context of hydrological research, our study contributes valuable insights to the body of literature on forecast enhancement techniques. Our findings can pave the way for more objective, on-the-fly selective forecasting system, tailored to catchment characteristics, time of year, and lead time, which would be a step towards user-friendly and practical hydrological forecasting systems.

In conclusion, this research provides a novel intercomparison of QM BC and PF DA, offering an assessment of their strengths and limitations when applied to UK streamflow forecasting. By recognising the diverse contexts where each method excels, hydrologists and decision-makers can make informed choices to refine forecasting products, aligning with the ever-growing demand for reliable and actionable hydrological information.



**Acknowledgements**

This work was undertaken through the Hydro-JULES research programme (NE/S017380/1), CANARI project
(NE/W004984/1), NC International programme (NE/X006247/1) and UK-SCAPE programme (NE/R016429/1) of the UK
Natural Environment Research Council (NERC). The authors would also like to thank Matthieu Chevallier at ECMWF who
carried out an internal review and provided valuable feedback on the paper. During the preparation of this paper the authors
used generative AI (ChatGPT) with the exclusive aim of enhancing readability and language. After using this tool, the authors
reviewed and edited the content as needed and take full responsibility for the content of the publication.

**Code and data availability**

All code used in this study was based on open-source libraries:

- Hydrological model GR4J: R package airGR (https://cran.r-project.org/web/packages/airGR/index.html)
- Verification: R package Easyverification (https://cran.r-project.org/web/packages/easyVerification/index.html)

The data used in this study is also from open-source datasets:

- River flow from the NRFA: https://nrfa.ceh.ac.uk/data
- Precipitation from GEAR dataset: downloadable at https://catalogue.ceh.ac.uk/documents/dbf13dd5-90cd-457a-a986-f2f9dd97e93c
- Potential evapotranspiration from CHESS for Great Britain and Historical-PET for Northern Ireland: downloadable at https://catalogue.ceh.ac.uk/documents/9116e565-2c0a-455b-9c68-558fdd9179ad and https://catalogue.ceh.ac.uk/documents/17b9c4f7-1c30-4b6f-b2fe-f7780159939c

**Author contributions**

MT led the overall analysis, experimental design, manuscript preparation, and supervised the team. ME led the Data
Assimilation analysis and contributed to manuscript preparation. AC conducted bias correction analysis, supervised junior
staff, and contributed to the manuscript. EM performed analyses to compare models and approaches, and produced figures for
the manuscript. EC supervised the Data Assimilation work and contributed to the manuscript. RJ developed code and
conducted bias correction analysis. KFC contributed to the experimental design, supervision, and manuscript preparation. JH
secured funding, supervised the work, and contributed to the experimental design and manuscript preparation.

**Competing interests**

The authors declare that they have no known competing financial interests or personal relationships that could have appeared
to influence the work reported in this paper.





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



**Appendix: Additional figures**

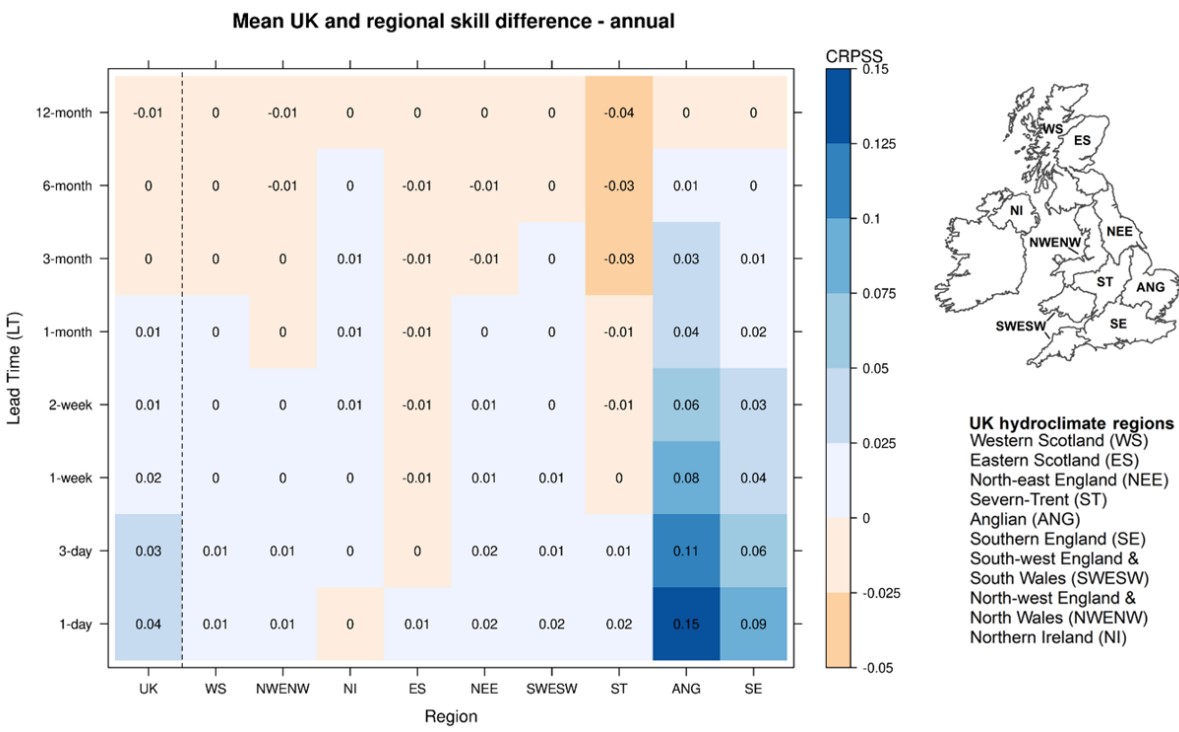


**Figure A1: Change in forecasting skill (CRPSS) at different lead time when transitioning from GR4J to GR6J to produce ESP forecasts in the different UK hydroclimate regions. Blue signifies improved forecast skill with GR6J compared to GR4J, while orange shades represent the reverse.**






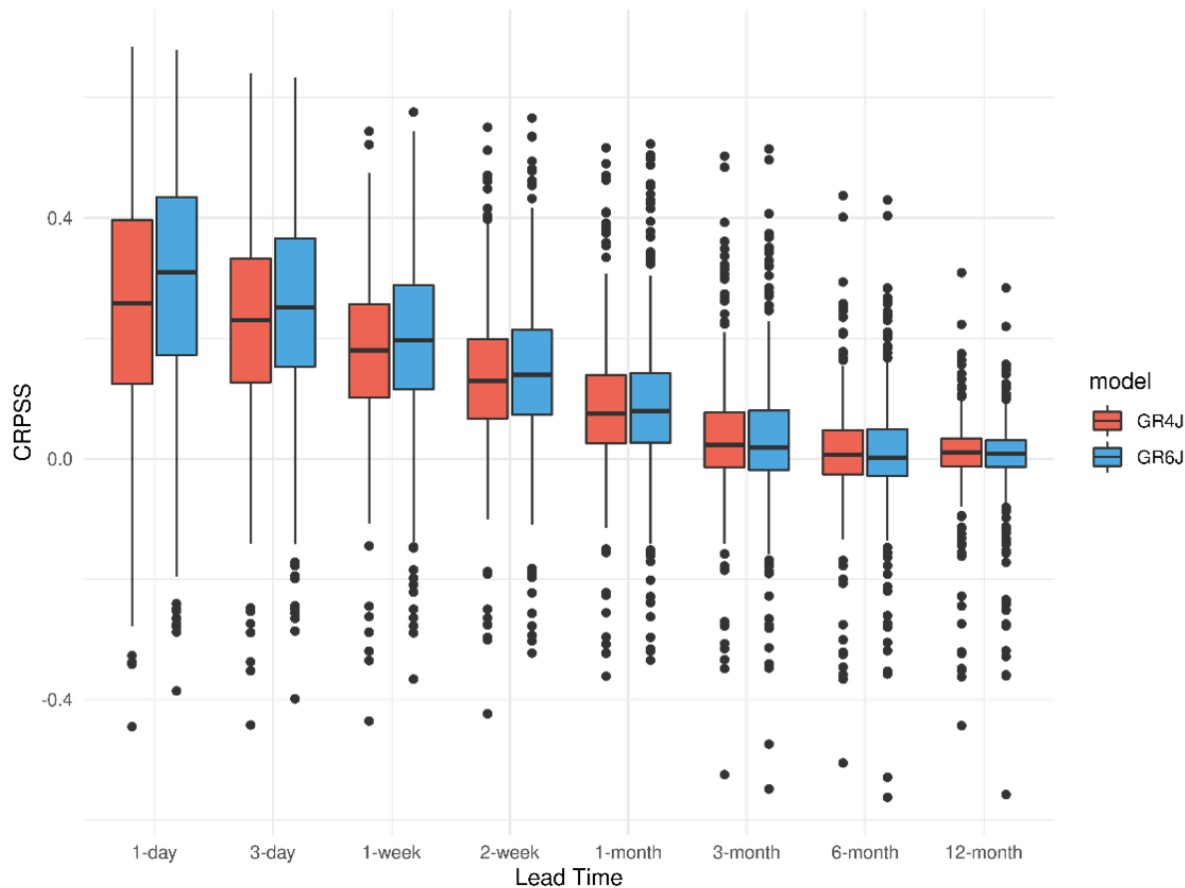

**Figure A2: Boxplots showing the range of CRPSS for all catchments used in the UKHO at different lead times in ESP forecasts generated using GR4J (in red) and GR6J (in blue).**






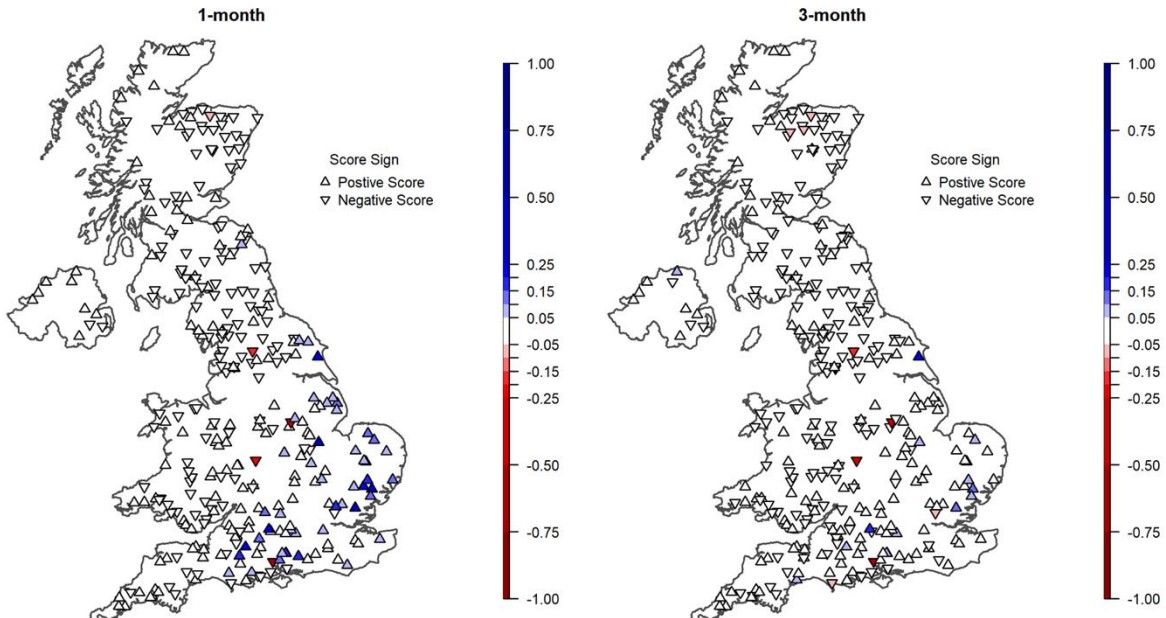

**Figure A3: Map of difference in skill (CRPSS) between ESP forecasts generated using GR6J and GR4J at (left) 1-month lead time and (right) 3-month lead time. Blue shades signify improved forecast skill with GR6J compared to GR4J, red shades represent the reverse, while white signifies negligeable differences.**








**Figure A4: Percent bias for each season for low flows (Q95) and high flows (Q05) in streamflow simulated by GR4J.**





**Figure A5: Bias (m³ s⁻¹) for each season for low flows (Q95) and high flows (Q05) in streamflow simulated by GR4J.**



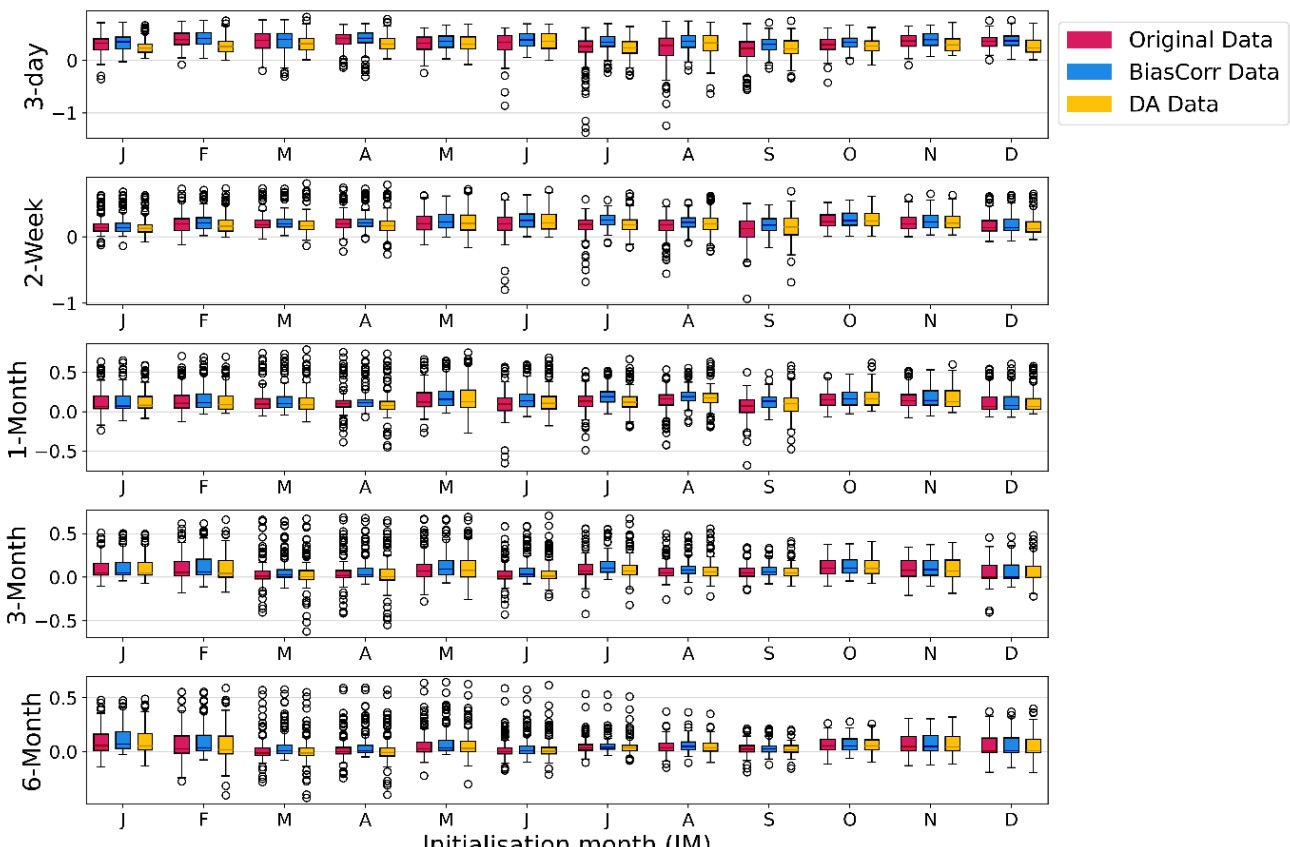

**Figure A6: CRPSS of OR-ESP, BC-ESP and DA-ESP forecasts at different lead times and initialisation months, for catchments with <5% of missing data.**




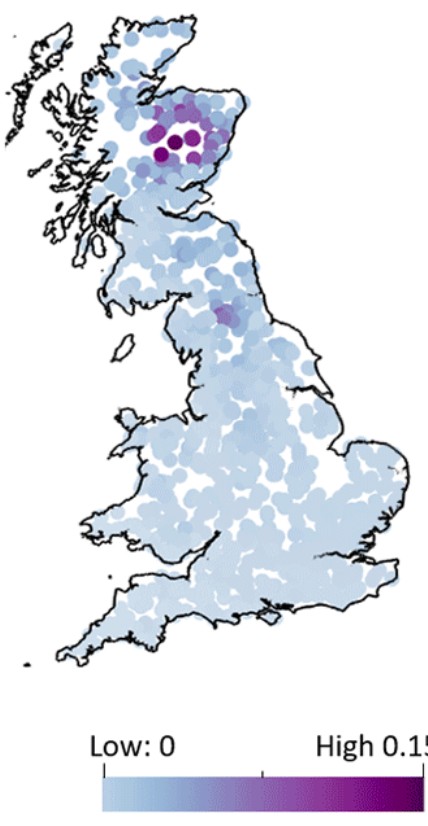

**Figure A7: Fraction of precipitation falling as snow for catchments across Great Britain, where a value of 0.15 indicates that 15 % of the catchment precipitation falls on days when the temperature is below zero (source: Lane et al., 2019)**
