# Peer review of "Optimising Ensemble Streamflow Predictions with Bias-Correction and Data Assimilation Techniques"

_Hydrology and Earth System Sciences, 2024_

## Referee Comment (RC1)

**Review of "Optimising Ensemble Streamflow Predictions with Bias-Correction and Data Assimilation Techniques" by Maliko Tanguy and collaborators.**

| Principal criteria | Excellent (1) | Good (2) | Fair (3) | Poor (4) |
|---|:---:|:---:|:---:|:---:|
| **Scientific significance**: Does the manuscript represent a substantial contribution to scientific progress within the scope of Hydrology and Earth System Sciences (substantial new concepts, ideas, methods, or data)? | | X | | |
| **Scientific quality**: Are the scientific approach and applied methods valid? Are the results discussed in an appropriate and balanced way (consideration of related work, including appropriate references)? | X | | | |
| **Presentation quality:** Are the scientific results and conclusions presented in a clear, concise, and well-structured way (number and quality of figures/tables, appropriate use of English language)? | X | | | |

Thank you for the opportunity to review this manuscript. I fully support the body of work presented in the Hydrological Outlook UK (HOUK), which offers a novel, brave, transformative, and impactful approach to comparing bias-correction (BC) and data assimilation (DA) techniques for improving hydrological forecasts using the Ensemble Streamflow Prediction (ESP) method in the UK. I applaud the authors for their significant accomplishments in this field.

For full disclosure, I acknowledge a potential bias in my review as I am the author of the Flow Duration Curve Quantile Mapping Bias Correction Method used in this study. I am honored that this method is contributing to operational hydrological forecasting.

In my view, this paper holds great value for publication in *Hydrology and Earth System Sciences (HESS)*. Based on my reading and review, I believe the manuscript would benefit from some "minor revisions." I recommend a few amplifications to enhance clarity for the reader. My specific comments are provided in the order they appear in the document.

In Line 48 or Line 55, it would be helpful to include a brief description of how the operational ESP currently works. For example, in 2024, which historical years are used to generate forecasts? Additionally, how are the initial hydrological conditions (IHCs) calculated for each

month in the operational setting? I suggest also including a comparison with the methodology used in this paper to highlight any key differences

In Line 85, Figure 1a is referenced, showing the absolute percent bias (absPBIAS) in streamflow simulations using the GR4J model. Could you clarify whether the absPBIAS data presented in this figure is based on results from this study or if it references a previous study, such as Smith et al. (2019)? If the figure represents data from the current study, it may be more appropriate to introduce or elaborate on it in the Results section to align with the manuscript's flow.

In Line 86, you reference Figures A4 and A5 in the appendix. While the figures provide valuable insights, the specific types of biases (e.g., percent bias, raw bias) illustrated in these figures have not been clearly defined. To enhance clarity, I suggest adding brief definitions or explanations of these biases directly in the appendix where Figures A4 and A5 are presented. This would help readers better understand the data without needing to refer back to earlier sections of the manuscript.

The following point is philosophical, and the authors are free to disagree or rebut, but I would appreciate your consideration. In the paragraph from Lines 96 to 108, you introduce the concept of Quantile Mapping Bias Correction (QM-BC). While Quantile Mapping (QM) typically involves adjusting the cumulative distribution function (CDF) of simulated data to match that of observed data, the method you describe uses Flow Duration Curves (FDCs) instead of CDFs. FDCs and CDFs, while similar in that they both involve sorting data and calculating probabilities, serve different purposes and represent different types of probabilities. FDCs focus on the exceedance probability of flow rates, making them particularly valuable in hydrological studies, whereas CDFs provide a broader statistical tool for analyzing any data distribution. Given this distinction, the bias correction method you are using might be more accurately described as a 'Flow Duration Curve Quantile Mapping Bias Correction Method (FDCQM-BC).' I suggest considering this terminology in the revised version of the paragraph to better align with the methodology you are applying. It's also important to note that some of the papers you cite (e.g., Chevuturi et al., 2023; Farmer et al., 2018) refer specifically to FDCQM-BC, while others (e.g., Usman et al., 2022; Li et al. 2017; Hashino et al., 2007; Wood and Schaake, 2008) discuss the typical QM method.

I would appreciate some clarification in section 2.2.2 on how the different forecasts are established. Here are the specific elements I would like to understand better

1. **Forecast Initialization and Horizon:**
   My understanding is that after running the hydrological model to obtain the 'Simulated Observed River Flows' dataset, the first three years are used to warm up the model. The OR-ESP is then calculated each month with this data (initialized on the 1st day of the month obtained from the 'Simulated Observed River Flows') and with the calibrated parameters with a 1-year forecast horizon. For example, the first forecast, starting on 1964-01-01, extends to 1964-12-31, the second forecast, starting on 1964-02-01, extends to 1965-01-31, and so on, until the last forecast starting on 2015-12-01 and

extending to 2016-11-30. Based on this, it seems there would be a total of 624 forecasts over this period. Could you confirm if this interpretation is correct? Additionally, I would appreciate an explanation that defines these details in the forecast datasets.

2. **Number of Ensembles in ESP Datasets:**
   The different ESP datasets (OR-ESP, BC-ESP, DA-ESP) contain 51 ensembles. Considering that these datasets span from 1964 to 2015. Using historical climate sequences from 1961 to 2015, I would expect 55 years of data. After applying the leave-three-years-out cross-validation (L3OCV), this would leave 52 ensembles. On the other hand, using historical climate sequences from 1964 to 2015, I would expect 52 years of data. After applying the leave-three-years-out cross-validation (L3OCV), this would leave 49 ensembles. Could you clarify how the number 51 was determined? Specifically, was either 1961 or 2015 excluded in generating the ESP data?

3. **Construction of Time Series for CRPSS:**
   I am unclear on how the time series used to calculate the Continuous Ranked Probability Skill Score (CRPSS) were constructed, especially for different forecast horizons like 1-day, 3-day, 1-week, etc. For example, assuming a 4-day forecast which is launched daily, we can build time series for Initialization, 1-day, 2-day, 3-day, and 4-day forecast as shown in the schema below:

| Dates | Forecast Starting on Jan. 1 | Forecast Starting on Jan. 2 | Forecast Starting on Jan. 3 |
|-------|-----------------------------|-----------------------------|-----------------------------|
| 1 - 1 | 11.2 cfs | | |
| 1 - 2 | 15.3 cfs | 15.5 cfs | |
| 1 - 3 | 19.7 cfs | 18.3 cfs | 18.1 cfs |
| 1 - 4 | 22.4 cfs | 23.7 cfs | 22.9 cfs |
| 1 - 5 | 10.9 cfs | 15.1 cfs | 16.7 cfs |
| 1 - 6 | | 13.1 cfs | 14.5 cfs |
| 1 - 7 | | | 11.2 cfs |

| Initialization Values (Water Balance) | | One Day Forecasts | | Two Day Forecasts | | Three Day Forecasts | | Four Day Forecasts | |
|---|---|---|---|---|---|---|---|---|---|
| 1 - 1 | 11.2 cfs | 1 - 2 | 15.3 cfs | 1 - 3 | 19.7 cfs | 1 - 4 | 22.4 cfs | 1 - 5 | 10.9 cfs |
| 1 - 2 | 15.5 cfs | 1 - 3 | 18.3 cfs | 1 - 4 | 23.7 cfs | 1 - 5 | 15.1 cfs | 1 - 6 | 13.1 cfs |
| 1 - 3 | 18.1 cfs | 1 - 4 | 22.9 cfs | 1 - 5 | 16.7 cfs | 1 - 6 | 14.5 cfs | 1 - 7 | 11.2 cfs |

I would appreciate a detailed description of how these time series were built. It may also be helpful to include more than one graphical schema to clarify this process. From my understanding, the 6-month forecast will include time series starting on 1964-06-01 (from the forecast launched on 1964-01-01) to 2016-05-31 (from the forecast launched on 2015-12-31 and would also include the months from 1964-07 (from the forecast launched on 1964-02-01), 1964-08 (from the forecast launched on 1964-03-01), and so on, until the last forecast month in May 2016. I assume, in a similar way, the time series for 30-day, 1-month, and 3-months were built. However I am not certain if this applies 1-day, 3-day, 1-week, and 2-week forecast horizon.

Additionally, could you clarify what the 'initialization month' in Figure 6 refers to? Does it represent the month when the forecast was launched, the start of the forecast time

series, or the specific month within the forecast time series data? Does this change the forecast time series construction?

I would appreciate some clarification in Section 2.4 regarding the Data Assimilation process. My understanding is that each time a new forecast is launched, the hydrological model needs to be adjusted to accurately reflect the initial hydrological conditions (IHCs). You mention that the Particle Filter (PF) method is used, which simulates potential scenarios with different sets of parameters. Once new observed data becomes available, the set of parameters most likely to describe the initial state of the forecast is determined, and these parameters are then used to run the model with the corresponding 51 historic sequences to calculate the DA-ESP. Could you clarify the following:

1. Is the period for applying the Particle Filter 4 years prior to the forecast launch? Is this understanding, correct?
2. Is there any specific error metric used to determine the new model parameters during the Data Assimilation process? If so, could you elaborate on which metrics are used and how they influence the selection of parameters?
3. If as supposed in the previous point, 624 forecasts were launched, it means 624 adjustments for initial conditions were made. Then, how the Data Assimilated (DA) 'Simulated Observed River Flows' time series is calculated?

In Table 2, Section 2.5, if Figure 1a corresponds to the results obtained in this study and includes the absPBIAS metric, I suggest adding absPBIAS to the list of performance metrics in the table.

In Table2, Section 2.5 the metric bias ratio $\beta = \dfrac{\mu_{\sqrt{q}}}{\mu_{\sqrt{Q}}}$ is presented. My understanding is that this ratio cannot be negative; it should range from 0 to $+\infty$, with 1 being the optimum value. However, in Figures A4 and A5, it appears that negative bias values are presented. Could you clarify whether the metric in these figures refers to this bias ratio ($\beta$), percent bias (Pbias), or another bias metric? How do the negative values arise in these figures if they are indeed based on the bias ratio?

In Section 2.6, I would appreciate it if you could include the mathematical formulation for the CRPSS calculation, detailing both the Continuous Ranked Probability Score (CRPS) and the skill score. This would help in understanding the specific methodology used for evaluating forecast skill. Additionally, I would like more details regarding the sentence: *'The Ferro et al. (2008) ensemble size correction for CRPS was applied to account for differences between the number of members in the hindcasts (51 members, corresponding to the historic period from 1961-2015 with the L3OCV approach) and the benchmark (47 members, corresponding to the period of 1965-2015 with the L3OCV approach and four years removed for the spin-up period), as done in the evaluation of hydrological ensemble forecasting elsewhere (e.g., Crochemore et al., 2017).'* While I understand the concept of hindcasts, the benchmarks are being mentioned here for the first time.  Furthermore, in Line 280, it is mentioned that the performance metrics from Section 2.5 were calculated for OR-ESP, BC-

ESP, and DC-ESP. However, these metrics do not appear in the results (as Section 3.1 seems to focus on 'Simulated Observed River Flows'). Similarly, Lines 282-284 reference additional metrics such as MAESS and MSESS, but these are also not shown in the results. Would it be possible to include these results in supplementary material? This would allow interested readers to explore these metrics further.

In Section 3.1, could you clarify whether the discussion is related to the 'Simulated observed river flows'? If it is, the process of obtaining the BC dataset is clear. However, I am unsure how the DA was calculated. From my understanding, 624 simulations were adjusted to fit the IHCs before each forecast launch, and with the 4-year spin-up period for initializing every forecast, wouldn't there be overlapping periods? Could you clarify how this was handled? If Section 3.1 is not related to the 'Simulated observed river flows' and instead pertains to OR-ESP, could you specify which forecast lead time was used? Additionally, was the mean ensemble used to calculate the performance metrics?

The following point is philosophical, and the authors are free to disagree or rebut, but I would appreciate your consideration. I suggest that Sections 3.2 and 3.3 could be rewritten to explain Figures 4 and 5 independently. While the results descriptions are clear, the current structure can be a bit confusing as it requires the reader to frequently refer back to the plots.

In Figure 6, I recommend placing the legend at the bottom to allow for a wider and taller plot, which would make it easier to see more details. Additionally, as mentioned earlier, could you clarify what the term 'initialization month' in Figure 6 refers to? Does it represent the month when the forecast was launched, the start of the forecast time series, or the specific month within the forecast time series data?

Lines 373-375 "*It is also interesting to note that there are cases where OR-ESP is better than both DA-ESP and BC-ESP (magenta points in Figure 7), especially in autumn, winter and beginning of spring (October to March) in the western part of the country for short lead-times (Figure 7a); and in spring for longer lead-times (Figure 7b) with no clear spatial pattern*." Could you provide any hypotheses or insights into why this situation might occur?

Personally, I loved the discussion and conclusions. I found them to be completely clear, precise, and pertinent

Thank you once again for the opportunity to review this manuscript. I hope the authors find my comments helpful, and I appreciate their understanding if any of my suggestions stem from a misunderstanding on my part

---

## Author Response (AR1)

**Optimising Ensemble Streamflow Predictions with Bias-Correction and Data Assimilation Techniques: Response to reviewers**

Maliko Tanguy, Michael Eastman, Amulya Chevuturi, Eugene Magee, Elizabeth Cooper, Robert H. B. Johnson, Katie Facer-Childs, and Jamie Hannaford

**Authors response to comments from reviewer #1:**

We would like to thank the reviewer for the very thorough review and positive feedback on our work. We appreciate the valuable suggestions and comments, which will significantly contribute to improving the clarity of our paper. Below, we have provided a detailed response (in blue) to each individual comment from the reviewer (in black). Note that the line numbers indicated in our responses corresponds to the marked-up version of the revised manuscript.

Thank you for the opportunity to review this manuscript. I fully support the body of work presented in the Hydrological Outlook UK (HOUK), which offers a novel, brave, transformative, and impactful approach to comparing bias-correction (BC) and data assimilation (DA) techniques for improving hydrological forecasts using the Ensemble Streamflow Prediction (ESP) method in the UK. I applaud the authors for their significant accomplishments in this field.
For full disclosure, I acknowledge a potential bias in my review as I am the author of the Flow Duration Curve Quantile Mapping Bias Correction Method used in this study. I am honored that this method is contributing to operational hydrological forecasting.

Thank you for your kind words and for recognising the value of our work. We also appreciate your Flow Duration Curve Quantile Mapping Bias Correction Method, which played an important role in our study and in advancing operational hydrological forecasting.

In my view, this paper holds great value for publication in Hydrology and Earth System Sciences (HESS). Based on my reading and review, I believe the manuscript would benefit from some "minor revisions." I recommend a few amplifications to enhance clarity for the reader. My specific comments are provided in the order they appear in the document.
In Line 48 or Line 55, it would be helpful to include a brief description of how the operational ESP currently works. For example, in 2024, which historical years are used to generate forecasts? Additionally, how are the initial hydrological conditions (IHCs) calculated for each month in the operational setting? I suggest also including a comparison with the methodology used in this paper to highlight any key differences.

Thank you for identifying this area for improvement. In the revised manuscript, we have provided additional details on the operational ESP and highlight key differences with the methodology used in this paper. We have added the following two paragraphs:

Line 45-51: "*The operational ESP uses all available years of historical meteorological data from 1961 onwards to generate forecasts, currently the period 1961–2024. Each year, a new ensemble is added as more data becomes available. The initial hydrological conditions (IHCs) are calculated by driving the hydrological model (currently GR6J; previously GR4J until November 2023) with observed meteorological data from the UK Met Office (UKMO) in near-real time. This data includes provisional, non-quality controlled precipitation and temperature grids (HadUK; Hollis et al., 2019). Potential evapotranspiration (PET) is calculated using the calibrated McGuinness-Bordne equation, as outlined by Tanguy et al. (2018).*"

Line 230-237: "*The main differences between the operational ESP and experimental set-ups in this paper are: (i) the ESP in the experimental set-up is constructed from 54 years of historical meteorological data (1961–2014), whereas the operational ESP currently uses 1961-2024, with a new ensemble added every year; (ii) the GR4J model is used in our analysis, as it was the operational model at the time, whereas the operational ESP uses GR6J since November 2023; and (iii) in our experiments, PET is derived from the Penman-Monteith based CHESS-PE, except in Northern Ireland, where McGuinness-Bordne PET was used due to data availability. For the operational ESP, McGuinness-Bordne PET is used over the whole of the UK. Since the McGuinness-Bordne equation is calibrated against CHESS-PE, we do not expect significant biases between the two PET calculation methods.*"

New References added to the manuscript:
*Hollis D, McCarthy MP, Kendon M, Legg T, Simpson I. HadUK-Grid—A new UK dataset of gridded climate observations. Geosci Data J. 2019; 6: 151–159.*
*https://doi.org/10.1002/gdj3.78*
*Tanguy, M., Prudhomme, C., Smith, K., and Hannaford, J.: Historical gridded reconstruction of potential evapotranspiration for the UK, Earth Syst. Sci. Data, 10, 951–968,*
*https://doi.org/10.5194/essd-10-951-2018, 2018.*

In Line 85, Figure 1a is referenced, showing the absolute percent bias (absPBIAS) in streamflow simulations using the GR4J model. Could you clarify whether the absPBIAS data presented in this figure is based on results from this study or if it references a previous study, such as Smith et al. (2019)? If the figure represents data from the current study, it may be more appropriate to introduce or elaborate on it in the Results section to align with the manuscript's flow.

The absPBIAS presented in Figure 1a is based on the analysis conducted for this study, and not from a previous study (although the results are indeed similar to those reported by Smith et al., 2019). We included this figure in the Introduction to justify the need for bias-correction in hydrological forecasts by emphasising the presence of bias in streamflow simulations early in the manuscript.

We acknowledge the reviewer's suggestion to place this in the Results section; however, we felt it was important to establish this motivation upfront as part of the justification. To avoid any confusion, we have explicitly clarified in the revised manuscript that the absPBIAS is based on the current study (in Figure1 caption).

In Line 86, you reference Figures A4 and A5 in the appendix. While the figures provide valuable insights, the specific types of biases (e.g., percent bias, raw bias) illustrated in these figures have not been clearly defined. To enhance clarity, I suggest adding brief definitions or explanations of these biases directly in the appendix where Figures A4 and A5 are presented. This would help readers better understand the data without needing to refer back to earlier sections of the manuscript.

Thank you for the suggestion. We have added the definition of the specific types of biases in the caption of the Figures in the revised version.

The following point is philosophical, and the authors are free to disagree or rebut, but I would appreciate your consideration. In the paragraph from Lines 96 to 108, you introduce the concept of Quantile Mapping Bias Correction (QM-BC). While Quantile Mapping (QM) typically involves adjusting the cumulative distribution function (CDF) of simulated data to match that of observed data, the method you describe uses Flow Duration Curves (FDCs) instead of CDFs. FDCs and CDFs, while similar in that they both involve sorting data and calculating probabilities, serve different purposes and represent different types of probabilities. FDCs focus on the exceedance probability of flow rates, making them particularly valuable in hydrological studies, whereas CDFs provide a broader statistical tool for analyzing any data distribution. Given this distinction, the bias correction method you are using might be more accurately described as a 'Flow Duration Curve Quantile Mapping Bias Correction Method (FDCQM-BC).' I suggest considering this terminology in the revised version of the paragraph to better align with the methodology you are applying. It's also important to note that some of the papers you cite (e.g., Chevuturi et al., 2023; Farmer et al., 2018) refer specifically to FDCQM-BC, while others (e.g., Usman et al., 2022; Li et al. 2017; Hashino et al., 2007; Wood and Schaake, 2008) discuss the typical QM method.

Thank you for your thoughtful comment regarding the terminology and conceptual distinctions in our manuscript. We recognise that Quantile Mapping (QM) can be applied using various distribution functions. While FDCs are indeed a specialised tool in hydrology, we consider the use of FDCs in this context as a specific implementation within the broader QM framework.

Our intention in using the term 'Quantile Mapping' was to maintain consistency with the broader QM literature, as FDCs can be viewed as an extension of the general QM approach tailored for hydrological data. We appreciate your suggestion and have clarified in the manuscript that our method applies FDCs within the QM framework (line 111-114).

I would appreciate some clarification in section 2.2.2 on how the different forecasts are established. Here are the specific elements I would like to understand better

**1. Forecast Initialization and Horizon:**

My understanding is that after running the hydrological model to obtain the 'Simulated Observed River Flows' dataset, the first three years are used to warm up the model. The OR-ESP is then calculated each month with this data (initialized on the 1st day of the month

obtained from the 'Simulated Observed River Flows') and with the calibrated parameters with a 1-year forecast horizon. For example, the first forecast, starting on 1964-01-01, extends to 1964-12-31, the second forecast, starting on 1964-02-01, extends to 1965-01-31, and so on, until the last forecast starting on 2015-12-01 and extends to 2016-11-30. Based on this, it seems there would be a total of 624 forecasts over this period. Could you confirm if this interpretation is correct? Additionally, I would appreciate an explanation that defines these details in the forecast datasets.

This interpretation is almost correct, but we realise that the explanation in the manuscript may have led to some misunderstanding. The last forecast actually starts on 2014-12-01, not 2015-12-01. At the time of analysis, both observed and simulated observed data were only available up to the end of 2015. Therefore, to verify the forecasts, the last forecast needed to finish within 2015. We did not include the forecast initialised on 2015-01-01, even though it ends on 2015-12-31, to maintain consistency; otherwise, the forecasts initialised in January would have had one additional instance compared to those from other initialisation months. As a result, there are a total of 612 forecasts, not 624. This has now been clarified in the revised manuscript (Line 215-229).

**2.  Number of Ensembles in ESP Datasets:**

The different ESP datasets (OR-ESP, BC-ESP, DA-ESP) contain 51 ensembles. Considering that these datasets span from 1964 to 2015. Using historical climate sequences from 1961 to 2015, I would expect 55 years of data. After applying the leave-three-years-out cross-validation (L3OCV), this would leave 52 ensembles. On the other hand, using historical climate sequences from 1964 to 2015, I would expect 52 years of data. After applying the leave-three-years-out cross-validation (L3OCV), this would leave 49 ensembles. Could you clarify how the number 51 was determined? Specifically, was either 1961 or 2015 excluded from generating the ESP data?

Thank you for your insightful comment. The confusion arises partly because the last forecast is initialised on 2014-12-01, making our statement in the manuscript that 2015 is included misleading (it should be 2014). Historical data from 1961 to 2014 accounts for a total of **54 ensembles** (not 55). While we use 3 years for the warm-up period, the meteorological data from 1961 to 1963 are still included as ensemble members. Thus, the total number of ensemble members is not reduced by the warm-up period – the 3 years of warm-up reduce the total number of forecasts produced, but not the number of ensemble members within each forecast.

Furthermore, due to the leave-three-years-out cross-validation (L3OCV), we subtract 3 years, resulting in a final count of **51 ensembles**. We have clarified this in the revised manuscript. Thank you for bringing this to our attention.

**3.  Construction of Time Series for CRPSS:**

I am unclear on how the time series used to calculate the Continuous Ranked Probability

Skill Score (CRPSS) were constructed, especially for different forecast horizons like 1-day, 3-day, 1-week, etc. For example, assuming a 4-day forecast that is launched daily, we can build a time series for Initialization, 1-day, 2-day, 3-day, and 4-day forecasts as shown in the schema below: I would appreciate a detailed description of how these time series were built.

| Dates | Forecast Starting on Jan. 1 | Forecast Starting on Jan. 2 | Forecast Starting on Jan. 3 |
|---|---|---|---|
| 1 - 1 | 11.2 cfs | | |
| 1 - 2 | 15.3 cfs | 15.5 cfs | |
| 1 - 3 | 19.7 cfs | 18.3 cfs | 18.1 cfs |
| 1 - 4 | 22.4 cfs | 23.7 cfs | 22.9 cfs |
| 1 - 5 | 10.9 cfs | 15.1 cfs | 16.7 cfs |
| 1 - 6 | | 13.1 cfs | 14.5 cfs |
| 1 - 7 | | | 11.2 cfs |

| Initialization Values (Water Balance) | | One Day Forecasts | | Two Day Forecasts | | Three Day Forecasts | | Four Day Forecasts | |
|---|---|---|---|---|---|---|---|---|---|
| 1 - 1 | 11.2 cfs | 1 - 2 | 15.3 cfs | 1 - 3 | 19.7 cfs | 1 - 4 | 22.4 cfs | 1 - 5 | 10.9 cfs |
| 1 - 2 | 15.5 cfs | 1 - 3 | 18.3 cfs | 1 - 4 | 23.7 cfs | 1 - 5 | 15.1 cfs | 1 - 6 | 13.1 cfs |
| 1 - 3 | 18.1 cfs | 1 - 4 | 22.9 cfs | 1 - 5 | 16.7 cfs | 1 - 6 | 14.5 cfs | 1 - 7 | 11.2 cfs |

It may also be helpful to include more than one graphical schema to clarify this process. From my understanding, the 6-month forecast will include time series starting on 1964-06-01 (from the forecast launched on 1964-01-01) to 2016-05-31 (from the forecast launched on 2015-12-31 and would also include the months from 1964-07 (from the forecast launched on 1964-02-01), 1964-08 (from the forecast launched on 1964-03-01), and so on, until the last forecast month in May 2016. I assume, in a similar way, the time series for 30-day, 1-month, and 3-months were built. However, I am not certain if this applies to the 1-day, 3-day, 1-week, and 2-week forecast horizon.

We realise that the construction of time series for the CRPSS calculation may not have been explained clearly enough. Have now revised the manuscript to provide a more detailed description of this process.

In brief, the model forecast is initialised on the 1st day of each month, after which the model runs freely for a lead time of up to 365 days, producing a forecast for each subsequent day at progressively longer lead times. For that month, no further initialisations are performed beyond the 1st day. Thus, all lead times (e.g., 1-day, 3-day, 7-day) are calculated relative to the 1st of the month.

To clarify how lead-times were handled: the CRPSS was calculated based on the accumulated flow over the full forecast period, rather than point values on specific days. For example, for the 7-day forecast horizon, the skill score is based on the total flow accumulated over the first 7 days of the forecast, not the streamflow value at precisely day 7. This approach better reflects the overall forecast performance for each lead-time by considering the cumulative discharge over the given period.

Regarding the 6-month forecast, we would consider the accumulated 6-month flow (at the end of the full 6-month) to calculate the CRPSS rather than the value for the 6[th] month of

forecast (see Figure R1 for an illustration of this concept). The same principle applies to the 1-day, 3-day, 1-week, and 2-week forecast horizons, where the CRPSS was based on the total accumulated flow over the respective lead time forecast periods.

These explanations are added in section 2.6, line 322-329.

[Figure]

*Figure R1:* *Streamflow forecast (dark blue solid line) and cumulative streamflow forecast (red solid line) for \*Random\* station initialised for 01-Jan-2000 for 12-months lead-time. Black arrow demonstrates the 6-month forecast lead time, with light blue arrow showing the corresponding 6-month forecast value and purple arrow showing the corresponding 6-month cumulative forecast value. The time series was generated randomly for conceptual illustration only.*

Additionally, could you clarify what the 'initialization month' in Figure 6 refers to? Does it represent the month when the forecast was launched, the start of the forecast time series, or the specific month within the forecast time series data? Does this change the forecast time series construction?

The 'initialisation month' in Figure 6 refers to the month when the forecast was launched. This month marks the beginning of the forecast period, and it coincides with the start of the forecast time series data. Therefore, the initialisation month serves as both the launch date of the forecast and the first month included in the forecast time series. This explanation has now been added to the figure caption.

I would appreciate some clarification in Section 2.4 regarding the Data Assimilation process. My understanding is that each time a new forecast is launched, the hydrological model needs to be adjusted to accurately reflect the initial hydrological conditions (IHCs). You mention that the Particle Filter (PF) method is used, which simulates potential scenarios with different sets of parameters. Once new observed data becomes available, the set of parameters most likely to describe the initial state of the forecast is determined, and these parameters are then used to run the model with the corresponding 51 historic sequences to

calculate the DA-ESP. Could you clarify the following:

1.   Is the period for applying the Particle Filter 4 years prior to the forecast launch? Is this understanding, correct?
Yes, this is correct: the PF was applied once a day during the four-year spin-up period, i.e. using daily streamflow observations. This resulted in a better initial condition at the end of the spin-up period, i.e., the start of the forecast period. Crucially, it also adjusted the GR4J parameters to better reflect those observations used during the spin-up period.
This is specified in the line 247 (in original manuscript) "*We used a daily updating timestep in the model spin up period (4 years), in order to improve the IHCs for the seasonal forecast*".
We have made this clearer, however by changing the text slightly to "*We applied the particle filter approach on daily data throughout the model spin up period (4 years), in order to improve the IHCs for the seasonal forecast*" (line 274-275 in revised manuscript).

2.   Is there any specific error metric used to determine the new model parameters during the Data Assimilation process? If so, could you elaborate on which metrics are used and how they influence the selection of parameters?

The PF applies Bayes' rule to estimate the posterior distributions of states and parameters by selecting the particles (model realisations) which best match the observation. We have added the following sentence to the revised manuscript: "*A sequential importance sampling approach was used to assign weights to individual particle states according to their likelihoods. This method is explained in more detail in Piazzi, et al. 2021.*" (line 276-278)

3.   If as supposed in the previous point, 624 forecasts were launched, it means 624 adjustments for initial conditions were made. Then, how the Data Assimilated (DA) 'Simulated Observed River Flows' time series is calculated?

To generate the 'Simulated Observed River Flows', the PF was applied once a day during the full evaluation period (1964-2015), i.e. using daily streamflow observations. This has been clarified in the revised manuscript. Then the IHCs produced in this way at the start of each month was used to run our ESP forecasts (612, not 624, as explained earlier).

In Table 2, Section 2.5, if Figure 1a corresponds to the results obtained in this study and includes the absPBIAS metric, I suggest adding absPBIAS to the list of performance metrics in the table.

Thank you for the suggestion. We have now added absPBIAS to Table 2.

In Table 2, Section 2.5 the metric bias ratio $\beta = \mu\_Vq/\mu\_VQ$ is presented. My understanding is that this ratio cannot be negative; it should range from 0 to $+\infty$, with 1 being the optimum value. However, in Figures A4 and A5, it appears that negative bias values are presented. Could you clarify whether the metric in these figures refers to this bias ratio ($\beta$), percent bias (Pbias), or another bias metric? How do the negative values arise in these figures if they are indeed based on the bias ratio?

We have added the definitions of the metric represented to the figure caption for clarity. Figure A4 illustrates the percent bias, calculated as $(q - Q)/Q*100$, for the low flows (Q95) and high flows (Q05), while Figure A5 shows the raw bias, defined as $(q - Q)$, for the low flows (Q95) and high flows (Q05); where q is simulated flow and Q is observed flow. Both bias and percent bias values can be negative when simulated flow is lower than observed flow.

In Section 2.6, I would appreciate it if you could include the mathematical formulation for the CRPSS calculation, detailing both the Continuous Ranked Probability Score (CRPS) and the skill score. This would help in understanding the specific methodology used for evaluating forecast skill.

Thank you for your comment. We reference the formulations for the Continuous Ranked Probability Score (CRPS) and the Continuous Ranked Probability Skill Score (CRPSS) in the manuscript (Hersbach, 2000). While the mathematical formulations are lengthy and complex, we believe they are not essential for understanding the core concepts and results of our paper.

We are open to including these formulas in the Appendix if the reviewer or editor feels strongly about it. Alternatively, readers can refer to the cited reference for detailed information.

Additionally, I would like more details regarding the sentence: 'The Ferro et al. (2008) ensemble size correction for CRPS was applied to account for differences between the number of members in the hindcasts (51 members, corresponding to the historic period from 1961-2015 with the L3OCV approach) and the benchmark (47 members, corresponding to the period of 1965-2015 with the L3OCV approach and four years removed for the spin-up period), as done in the evaluation of hydrological ensemble forecasting elsewhere (e.g., Crochemore et al., 2017).' While I understand the concept of hindcasts, the benchmarks are being mentioned here for the first time.

Thank you for your comment. You are correct that the benchmark used for verification was not mentioned previously, which was an oversight on our part. In the revised manuscript, we have clarified that we used climatology as our benchmark for evaluation, as this is a common choice for assessing sub-seasonal to seasonal forecasts. Additionally, we will briefly mention other common benchmarks for context, such as the persistence forecast, where the current state is maintained constant for future predictions and is typically used for shorter-range forecasts, and the gain-based benchmark, where simpler models are employed to generate forecasts (Pappenberger et al., 2015). This has been added in line 330-335.

We do not plan to explain the Ferro et al. (2008) method for ensemble size correction in detail, as this method is readily available within the EasyVerification R package. Including this information could unnecessarily complicate the manuscript. If readers are interested in understanding the method, they can refer to the original paper cited.

Reference:

Pappenberger, F., Ramos, M.H., Cloke, H.L., Wetterhall, F., Alfieri, L., Bogner, A.K., Mueller, Salamon, P. (2015). How do I know if my forecasts are better? Using benchmarks in hydrological ensemble prediction, Journal of Hydrology, 522, 697-713, https://doi.org/10.1016/j.jhydrol.2015.01.024.

Furthermore, in Line 280, it is mentioned that the performance metrics from Section 2.5 were calculated for OR-ESP, BC-ESP, and DC-ESP. However, these metrics do not appear in the results (as Section 3.1 seems to focus on 'Simulated Observed River Flows'). Similarly, Lines 282-284 reference additional metrics such as MAESS and MSESS, but these are also not shown in the results. Would it be possible to include these results in supplementary material? This would allow interested readers to explore these metrics further.

Thank you for highlighting this issue. The confusion stems from the statement in Line 280; the performance metrics were indeed calculated for the 'simulated observed flows' rather than for the different versions of ESP. This has now been corrected in the revised manuscript.

Regarding the inclusion of MAESS and MSESS in the results, we believe that adding these metrics would not significantly enhance the value of the manuscript. The results for MAESS and MSESS are very similar to those obtained with CRPSS, and including them would not alter our conclusions. This aligns with the approach taken by Harrigan et al. (2018), who also calculated these metrics but chose to present only the CRPSS results due to the lack of significant differences observed across various metrics.

Furthermore, both MAESS and MSESS are deterministic skill scores, while our analysis is based on probabilistic forecasts. Therefore, we consider CRPSS to be the most appropriate metric for evaluating our forecast performance in this context.

We could consider removing the mention of these two metrics altogether from the manuscript to avoid confusion, but have decided to keep their mention in, as our intention was to acknowledge that different metrics yielded consistent results.

We include in this response the versions for MAESS and MSESS of Figures 4 to 6, but we have not included them in the revised version for the reasons stated above.

[Figure]

***Figure R2:*** *Same as Figure 4 but for MAESS*

[Figure]

***Figure R3:*** *Same as Figure 4 but for MSESS*

[Figure]

**Figure R4:** *Same as Figure 5 but for MAESS*

[Figure]

***Figure R5:*** *Same as Figure 5 but for MSESS*

[Figure]

***Figure R6:*** *Same as Figure 6 but for MAESS*

[Figure]

**Figure R7:** *Same as Figure 6 but for MSESS*

In Section 3.1, could you clarify whether the discussion is related to the 'Simulated observed river flows'? If it is, the process of obtaining the BC dataset is clear. However, I am unsure how the DA was calculated. From my understanding, 624 simulations were adjusted to fit the IHCs before each forecast launch, and with the 4-year spin-up period for initializing every forecast, wouldn't there be overlapping periods? Could you clarify how this was handled? If Section 3.1 is not related to the 'Simulated observed river flows' and instead pertains to OR-ESP, could you specify which forecast lead time was used? Additionally, was the mean ensemble used to calculate the performance metrics?

Section 3.1 is indeed related to the 'simulated observed river flows', we have made that clear in the revised manuscript.
For DA, as explained earlier, to generate the 'Simulated Observed River Flows', the PF was applied once a day during the full evaluation period (1964-2015), i.e. using daily streamflow observations. Then the IHCs produced in this way at the start of each month was used to run our ESP forecasts. This has been added to section 2.4.

The following point is philosophical, and the authors are free to disagree or rebut, but I would appreciate your consideration. I suggest that Sections 3.2 and 3.3 could be rewritten to explain Figures 4 and 5 independently. While the descriptions of the results are clear, the

current structure can be a bit confusing as it requires the reader to frequently refer back to the plots.

We appreciate your suggestion to enhance readability and recognise that the current structure may cause some confusion due to the frequent references between Figures 4 and 5. We agree that such cross-referencing can occasionally disrupt the flow of the narrative.

However, we believe that the current structure, which first outlines the overarching trends in Figure 4 and then provides more detailed breakdowns in Figure 5, first for DA, then for BC, offers the most natural and coherent presentation of the results. Figure 4 serves as a high-level summary of forecast skill improvements across different scenarios, while Figure 5 delves deeper into specific factors such as catchment type and seasonality. By structuring the discussion in this way, we aim to provide both a broad overview and in-depth insights, which would be difficult to achieve if the figures were presented entirely independently.

The results for DA and BC are shown as subfigures within the same figure to facilitate direct comparison between the two approaches. This arrangement allows readers to easily juxtapose the results side by side, but is also the reason why the figures are referenced in both Sections 3.2 and 3.3.

After careful consideration, we feel that, while this structure may require readers to refer back and forth between the figures, it still offers the clearest and most logically consistent flow of the narrative. As such, we have decided to retain the current structure.

In Figure 6, I recommend placing the legend at the bottom to allow for a wider and taller plot, which would make it easier to see more details.

Thank you for the suggestion. We have made that amendment in the revised version. This is the revised Figure 6:

[Figure]

**Figure R8:** *Revised Figure 6*

Additionally, as mentioned earlier, could you clarify what the term 'initialization month' in Figure 6 refers to? Does it represent the month when the forecast was launched, the start of the forecast time series, or the specific month within the forecast time series data?

It refers to both the month where the forecast was launched, and the start of the forecast time series. We have added this in the caption of the figure in the revised version.

Lines 373-375 "It is also interesting to note that there are cases where OR-ESP is better than both DA-ESP and BC-ESP (magenta points in Figure 7), especially in autumn, winter, and beginning of spring (October to March) in the western part of the country for short lead-times (Figure 7a); and in spring for longer lead-times (Figure 7b) with no clear spatial pattern." Could you provide any hypotheses or insights into why this situation might occur?

This is an interesting question. While we do not have a definitive explanation, we can offer some speculative insights into the results:

The western part of the UK is much wetter than the rest of the country. In such a wet climate, the initial conditions may play a less significant role in determining forecast accuracy, including at shorter lead times. Consequently, the performance of OR-ESP could occasionally appear superior to that of the corrected forecasts simply due to the inherent

variability of rainfall. If the rainfall predictions are significantly off, it could overshadow any advantages gained from data assimilation or bias correction, leading to situations where OR-ESP 'gets it right' by chance.

As for the longer lead times observed in spring, this season is a transition period, making accurate forecasting particularly challenging (Harrigan et al., 2018). Even with the benefits of bias correction or data assimilation, the unpredictability of this period could result in forecasts that lack real skill. Thus, there may be a higher likelihood of OR-ESP performing better simply by chance rather than reflecting true forecasting skill.

Although these hypotheses offer some potential explanations, we do not feel confident enough to include them in the manuscript without further validation.

Personally, I loved the discussion and conclusions. I found them to be completely clear, precise, and pertinent.

Many thanks for your positive feedback. We are pleased to know that you enjoyed our discussion and conclusion.

Thank you once again for the opportunity to review this manuscript. I hope the authors find my comments helpful, and I appreciate their understanding if any of my suggestions stem from a misunderstanding on my part.

Thank you again for your valuable review. Your feedback has been uplifting and very helpful. Your comments have been valuable in improving the clarity and quality of our work.

**Authors response to comments from reviewer #2:**

We are grateful to the reviewer for the insightful review and feedback on our work. The suggestions and comments have contributed to improving the quality of our paper. Below, we have provided a detailed response (in blue) to each individual comment from the reviewer (in black).

I thank the authors for an interesting and well-written manuscript. The paper describes a comparison of error correction methods - bias correction and data assimilation - as applied to ensemble streamflow prediction for a large set of hydrologic models and forecast locations throughout the UK. The motivations, objectives, method, and results were, in general, well described and easy to understand.
Furthermore, considerations related to, but arguably beyond the scope of a single study, were discussed in an honest and helpful manner. This is commendable and makes the manuscript overall more useful.
The manuscript is suitable for publication with some minor revisions. I provide some general review comments and some specific suggested edits below.

We thank the reviewer for their positive feedback. Please find our responses to your specific comments one-by-one below.

Much of the results and discussion presented relies on interpreting differences between two or three ensembles of results that, in many cases, have considerable overlap. It would be useful to understand whether these apparent differences (such as in Figure 3 and Figure 4) are meaningful. Have the authors considered any statistical tests to determine if any such differences should be considered significant?

We thank the reviewer for this pertinent comment. We acknowledge that some of the differences observed are small, with considerable overlap between methods, raising the question of whether these differences are statistically significant.

In response, in the revised version, we have applied paired t-tests to evaluate whether the differences in performance are meaningful. Specifically:

▪ **Figure 3:** We have applied a paired t-test to the performance metrics of the deterministic simulations across the 316 catchments to determine whether a significant number of stations show improvement with the application of BC or DA methods. We found that for almost all the performance metrics, the GR4J with BC and GR4J with DA differ significantly from GR4J model with no additional processing at the 5% significance level.

▪ **Figures 4 and 6:** Similarly, we have applied paired t-tests to assess whether the differences in CRPSS values (across different lead times and initialisation months) are statistically significant. This shows that, for BC, OR-ESP and BC-ESP differ significantly at the 5% significance level for almost all lead time (Figure 4c and 4d), and most initialisation months, especially for 1 to 3 months lead time (Figure 6). The difference between OR-ESP and DA-ESP is significant at the 5% significance level mostly for shorter

lead time (Figure 4a and 4b), and for summer and autumn initialisation months (Figure 6).

In all cases, we have compared the OR simulations with the DA and BC simulations to assess the overall improvements introduced by the new approaches relative to the original method. We have also carried out these significance testing using the Kolmogorov-Smirnov test, and these showed similar results (not shown).

A sentence has been added line 343-346 to describe the significance testing towards the end of section 2.6, and in the revised version of Fig. 3, 4 and 6, a cross or a star indicates whether the differences are significant or not (explanation added to the caption).

We also clarify that CRPSS is a probabilistic forecast skill score for the entire ensemble. For each station, initialisation month, and lead time, there is only one value for CRPSS, with the spread representing the variation across the 316 catchments. While the paired t-tests will provide insights into the overall effectiveness of the methods across catchments, it may not fully inform the best method at the station level. Such decisions should take into consideration not only improvements in accuracy but also resource availability and user-specific requirements (see Sections 4.1; 4.3 and Table 3).

With the data assimilation approach, it is not immediately clear how updating system state should improve accuracy in snow-dominated catchments if the GR4J models being used do not include a snow model component.

We thank the reviewer for this insightful comment. While it is true that GR4J does not explicitly model snow accumulation or melt processes, it has been shown that the ESP approach exhibits skill in snow-dominated catchments (Harrigan et al., 2018). This is likely due to the model's ability to capture hydrological dynamics in situations where snow accumulation and melt are not the dominant processes. Furthermore, the calibration of the unit hydrograph step sizes, as well as the routing and production store sizes, may help the model capture the lagged response to snowfall. Therefore, improving the estimation of their states could improve skill in snow-dominated catchments.

Is there a risk that the DA process actually introduces a process inconsistency, or is it expected that the 4-year spin-up period should include sufficient variability to smooth out any bias in model state (soil moisture or baseflow stores) caused by matching flow without the snow present? Some additional discussion of this would bolster the overall interpretation of the DA method.

As GR4J does not include a snow module, snowmelt and accumulation are likely captured indirectly through other processes in the model. The data assimilation process updates the model state, including these processes, which can improve the model's ability to simulate snow dynamics, even though they are not explicitly represented. Including a dedicated snow module would allow these processes to be controlled separately, offering a more direct representation.

Regarding the potential for process inconsistency, data assimilation can indeed implicitly capture physical processes not explicitly represented in the model, provided there is sufficient signal in the observations. For example, Cooper et al. (2021) acknowledge that the updated parameters in the JULES model are likely correcting for groundwater processes not included in the model. In a similar way, GR4J might implicitly account for snow-related processes through data assimilation, even though they are not explicitly included in the model.

As the reviewer correctly points out, it is important to use observations with and without snow during the spin-up period to ensure that seasonal variability is captured properly. A future improvement would be to explicitly include snow-related processes in the model. This could be tested by repeating the optimisation with a model that includes snow representation (e.g. GR4J-CemaNeige), which would allow us to compare updated parameters and assess how much the snow processes influence the parameter values. We have added a discussion of this issue in the revised manuscript (line 493-500).

**Reference:**
Cooper, E., Blyth, E., Cooper, H., Ellis, R., Pinnington, E., and Dadson, S. J.: Using data assimilation to optimize pedotransfer functions using field-scale in situ soil moisture observations, Hydrol. Earth Syst. Sci., 25, 2445–2458, https://doi.org/10.5194/hess-25-2445-2021, 2021.

I'm curious if the choice to conduct the bias correction on a monthly basis has any effect on the relative value it provides to forecasts at different lead times. Would a bias correction on a weekly, or bi-weekly basis improve short lead time forecasts? Some justification for the use of the monthly application of BC would aid overall understanding.

The decision to apply bias correction (BC) on a monthly basis was driven by the seasonal variability in the UK hydrological system (wet winters, drier weather and higher evaporative demand in summer, with spring and autumn being transition seasons). Monthly BC captures these seasonal changes while avoiding overfitting to short-term fluctuations.

While weekly or bi-weekly BC could improve short-term forecasts, the high variability inherent in the UK climate means that such frequent corrections might introduce noise rather than enhance accuracy. Monthly BC provides a balanced approach, adjusting for seasonality without overreacting to short-term extremes.

We have clarified this rationale in the revised manuscript (line 246-251).

Under the bias correction approach, what mechanism can be invoked to explain why the predictive ability is greater than climatology (i.e. a positive skill score)? It seems that if the models are being driven by historical meteorology, and errors in the models are reduced by bias correction, then the model results should approximate climatological streamflow - especially at longer lead times. Some additional explanation on this topic would clarify and bolster the bias correction component of the manuscript.

The greater predictive ability of the bias-corrected forecast, compared to climatology, can be attributed to the role of initial conditions in hydrological forecasting. The model is initialised using hydrological simulations driven by observed meteorological data up to the start of the forecast period. Since the model (GR4J) has been calibrated, it provides a good estimate of the initial conditions, which serves as a strong foundation for the forecast.

This good estimate of initial conditions, combined with the hydrological memory inherent in many hydrological systems, is what drives the skill of the forecast, even at longer lead times. In contrast, a climatological forecast does not adjust initial conditions, meaning it lacks the benefit of model-based initialisation. As a result, the forecast skill from the bias-corrected model, which incorporates both observed initial conditions and bias adjustment, is expected to outperform climatology, even when driven by historical meteorology.

We have added a more detailed explanation of this mechanism in the revised manuscript to clarify the benefit of the bias correction approach (line 423-429).

Page 1 - Table 1: A very minor comment: The glossary is much appreciated, but it seems a bit odd to start the manuscript with it. Perhaps in typesetting this could be moved to a location early in the manuscript but following some introductory text.

We thank the reviewer for this suggestion. We agree that the glossary could be better placed later in the manuscript, following some introductory text. We have now moved the glossary to the end of the introduction.

Page 9 Line 223 - I might suggest "satisfied" instead of the word "verified" here.

Corrected in revised manuscript.

Page 11 - Section 2.6: Provide a bit more explanation of the CRPSS to help the reader understand how results should be interpreted. An explanation of the calculation method, the scale (what do different potential score values mean [e.g. 0 means no skill, 1 perfect, etc]). Also, it's not necessarily clear on what basis the skill score is calculated at different lead times - is it cumulative volume, mean flow rate over the forecast period, or something different?

We thank the reviewer for this helpful comment. In the revised manuscript, we will expand our explanation of the Continuous Ranked Probability Skill Score (CRPSS) to clarify its calculation and interpretation.

The CRPSS measures the relative skill of the forecast compared to a benchmark, in this case, climatology. It is defined as:

$$CRPSS = 1 - \frac{CRPS_{forecast}}{CRPS_{climatology}}$$

where:

- CRPS$_{forecast}$ is the Continuous Ranked Probability Score (CRPS) of the forecast ensemble, calculated by comparing the Cumulative Distribution Function (CDF) of the forecast to the observed data over the evaluation period.
- CRPS$_{climatology}$ is the CRPS for the climatology (our benchmark), calculated by comparing the CDF of the climatology (our benchmark) to the observed data over the same period.

The CRPSS values are interpreted as follows:

- CRPSS=1: The forecast has perfect skill compared to climatology.
- CRPSS=0: The forecast has no skill compared to climatology (forecast is as good as using climatology).
- CRPSS<0: The forecast is less accurate than climatology (forecast is misleading, and has no skill).

The CRPSS is evaluated over cumulative flow at different lead times, reflecting how well the forecasted distribution aligns with observed cumulative flow. This method ensures a fair comparison of the skill of forecasts over the entire forecast horizon.

We have included this explanation in the revised manuscript (line 301-314) to clarify both the calculation of CRPS and the interpretation of CRPSS. Readers interested in further details on the methodology are already referred to Hersbach (2000) in the manuscript.

Line 271 - Should "CRPS" be "CRPSS" for consistency, or is this not referring to a skill score in this usage?

In this case, the term "CRPS" (Continuous Ranked Probability Score) is used correctly, as we are referring to the raw score rather than the skill score. As explained in the previous point, the CRPS is a standalone metric that compares the forecast's cumulative distribution to the observed outcome, without reference to a benchmark.

The CRPSS (Continuous Ranked Probability Skill Score), on the other hand, is derived from the CRPS by comparing the forecast's CRPS to that of a benchmark (in this case, climatology). This has now been clarified in the revised manuscript in response to the previous comment, which should hopefully help readers understand the difference between CRPS and CRPSS.

Figure 3: Top left subplot should be "rmse" not "rsme"

Noted, this has now been corrected in the revised manuscript.

Page 18 - Figure 7: Perhaps some missing words in the caption - consider revising.

Thank you for pointing that out. The caption has been revised to: *Each catchment showing the best performing method (OR-ESP: Red; BC-ESP: Blue and DA-ESP: Yellow) based on CRPSS for each month at lead time of (a) 3 days and (b) 30 days.*